# Pesticide peak concentration reduction in a small vegetated treatment system controlled by chemograph shape

Jan Greiwe[1], Oliver Olsson[2], Klaus Kümmerer[2], Jens Lange[1]

[1]Hydrology, University of Freiburg, Fahnenbergplatz, 79098 Freiburg, Germany

[2]Institute of Sustainable and Environmental Chemistry, Leuphana University of Lueneburg, Scharnhorststr. 1, 21335 Lueneburg, Germany

*Correspondence to*: Jan Greiwe (jan.greiwe@hydrology.uni-freiburg.de)

**Abstract.** Pesticides may impact aquatic ecosystems when entering water bodies. Measures for mitigation against pesticide inputs include vegetated treatment systems (VTS). Some of these systems have very short hydraulic retention time (< 1 h) but nevertheless manage to effectively reduce peak concentrations of contaminants as a result of dispersion. We hypothesize that the effect of dispersion on contaminant mitigation in VTS depends on the shape of the contaminant input signal chemograph which in turn is related to factors affecting contaminant mobilization in the contributing catchment. In order to test this hypothesis we grouped chemographs of 6 contaminants originating from a viticultural catchment during 10 discharge events into clusters according to chemograph shape. We then compared peak concentration reduction and mass removal in a downstream VTS both among clusters and in terms of compound properties and discharge dynamics. We found that chemograph clusters reflected combined effects of contaminant source areas, transport pathways, and discharge dynamics. While mass loss was subject to major uncertainties, peak concentration reduction rate was clearly related to chemograph clusters and dispersion sensitivity. These findings suggest that mitigation of acute toxicity in VTS is stronger for compounds with sharp-peaked chemographs whose formation is related to the contributing catchment and can be analyzed by chemograph clustering.

## 1    Introduction

Use of pesticides is beneficial for agricultural productivity. However, when pesticides reach surface water bodies, they threaten aquatic organisms (Zubrod et al., 2019). Effects of pesticides on aquatic ecosystems include a reduction of species richness of invertebrates (Beketov et al., 2013) as well as microorganisms (Fernández et al., 2015). Unintended export of pesticides from the application site to water bodies can happen in particulate form via erosion (Oliver et al., 2012; Taghavi et al., 2011) or in dissolved form via surface runoff, drainage pipes, spray drift or leakage to groundwater (Reichenberger et al., 2007) and subsequent exfiltration.

In the environment pesticides are subject to degradation by both abiotic (e.g. hydrolysis, photolysis) and biotic (e.g. plant metabolism, microbial degradation) processes (Fenner et al., 2013). If degradation is incomplete, pesticides form

transformation products (TPs) which in some cases may be equally or more mobile, persistent or toxic than their PCs (Hensen
et al., 2020). Strongest mobilization of pesticides is usually associated with the first significant rainfall after application and
fast discharge components (Doppler et al., 2012; Olsson et al., 2013; Lefrancq et al., 2017), e.g. runoff from non-target areas
like roads (Lefrancq et al., 2013). Mobilization dynamics of TPs usually differ from those of their parent compounds (PCs) in
terms of source areas and export pathways. The formation of TPs may happen on larger time scales and TPs usually have
different physicochemical properties than their PCs (Gassmann et al., 2013). The specific transformation and further
degradation of a contaminant largely depends on the interplay of the contaminant's mobility and degradability as well as site
characteristics (Gassmann et al., 2015). Both mobility and degradability can vary over multiple orders of magnitude for
different contaminants. However, water and soil half-lives are at least in the order of several days or weeks for most pesticides
(Lewis et al., 2016).
Pesticide pollution can be mitigated by vegetated treatment systems (VTS) located between source areas and receiving water
bodies (Vymazal and Březinová, 2015; Stehle et al., 2011; Gregoire et al., 2009). Such systems temporally retain polluted
waters and thus provide space, time and favorable conditions for degradation processes. VTSs studied in literature include
very different types of systems (Lange et al., 2011), including vegetated ditches or detention ponds with hydraulic residence
times (HRT) ranging in the order of minutes to several hours (Bundschuh et al., 2016; Elsaesser et al., 2011; Ramos et al.,
2019) or constructed wetlands in which HRT may reach several weeks (Maillard and Imfeld, 2014), particularly when operated
in batch mode (Tournebize et al., 2017; Maillard et al., 2016; Moore et al., 2000). The term pesticide mitigation can refer to
contaminant mass removal ($R_M$) or peak concentration reduction ($R_C$). While mass removal is mainly observed in systems with
longer HRT and affects permanent toxicity, peak concentration reduction also happens in systems with short HRT where it
reduces acute toxicity (Bundschuh et al., 2016; Elsaesser et al., 2011; Stehle et al., 2011).
During longitudinal transport in streams or wetlands, peak concentration reduction does not necessarily involve degradation
but can solely be the result of enhanced dispersion due to the presence of obstacles such as plants (Elsaesser et al., 2011) and
temporary removal from the water phase by reversible sorption. Mitigation properties therefore constantly change due to
wetland succession (Schuetz et al., 2012). Regardless of whether VTSs target concentration reduction or mass removal,
mitigation efficiency is usually associated with physicochemical properties of target compounds (Vymazal and Březinová,
2015) or VTSs, including their operation mode (Tournebize et al., 2017). However, following the concept of advective-
dispersive transport (Fischer et al., 1979), the mitigating effect of dispersion on a concentration signal does not only depend
on the magnitude of dispersion but also on the shape of the signal. Peak concentration reduction will be stronger for a signal
with a pronounced peak and low background than for a signal with a small peak and high background if both signals are
exposed to the same dispersion. Chemograph shapes, in turn, are dictated by processes in the contributing catchments. The
influence of this chain of effects on contaminant mitigation and hence VTS efficiency has not been systematically investigated
so far.
We hypothesize that the efficiency of contaminant mitigation in VTSs depends on the shape of the input chemographs and
eventually on the factors that produce these signals in the catchment. In order to test this hypothesis we grouped chemographs

of 6 contaminants mobilized in a viticultural catchment during 10 discharge events into clusters according to chemograph shape. We then compared peak concentration reduction and mass removal in a downstream VTS both among clusters and in terms of compound properties and discharge dynamics.

## 2    Material and methods

### 2.1    Study site

The study site (Figure 1) is located inside a flood detention basin in the 1.8 km² Loechernbach catchment, southwest Germany. Catchment elevation ranges between 213 m.a.s.l. at the outlet and 378 m.a.s.l. in the western corner. Mean precipitation was 800 mm a$^{-1}$ between 2009 and 2018, mean air temperature 11.3 °C. Soils mainly consist of calcaric regosols which formed on aeolian loess and have a typical grain size distribution of 10 % sand, 80 % silt and 10 % clay. Most of the catchment is dedicated to large artificial vineyard terraces (71 %), while croplands occupy the valley bottoms (20 %). Forest only accounts for a small portion (9 %) and is limited to the most elevated part of the catchment. This partition in land use is reflected in the main application areas of pesticide types. Fungicides are applied on vineyard terraces, while herbicides are mainly applied to the cropland in the flat valleys. Large parts of the catchment are drained by a sub-surface pipe network (Figure 1) connecting vineyards and paved roads to the main channel in the valley. This drainage network causes fast downstream transport of storm water and suspended sediments (Gassmann et al., 2012). In addition, fields in the valley bottoms are drained by a secondary network of smaller and usually shorter field drains that are either connected to the primary drainage network or directly connected to the stream (Schuetz et al., 2016). A 20,000 m$^3$ detention basin was built at the outlet of the Loechernbach to prevent flooding of the downstream village.Inside the detention basin, a 258 m² vegetated surface flow constructed wetland and a 105 m² retention pond (maximum depth 1.5 m) are connected in series parallel to the course of the Loechernbach stream. A small dam diverts all flow through the vegetated treatment systems during base flow conditions, but allows water to bybass the VTS during large discharge events. The wetland is in operation since 2010 and its succession was studied by Schuetz et al. (2012). The pond was added to the system in January 2016. The entire detention basin is sealed by an impermeable clay layer that prevents leakage to groundwater. Water residence times range from less than one hour during flood events up to several days during extreme low flow conditions.

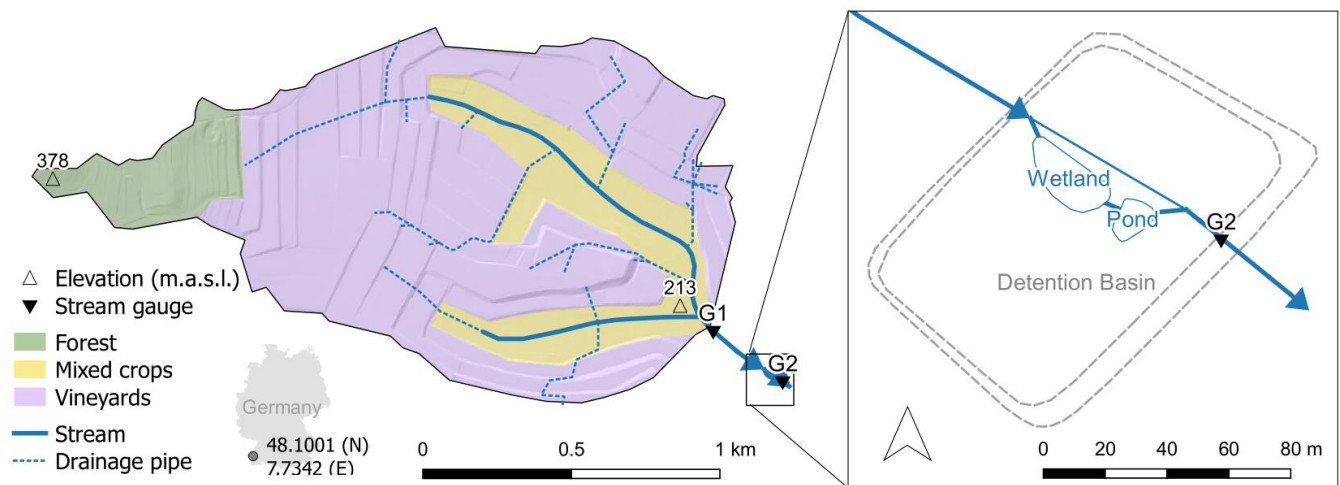

Figure 1: Loechernbach catchment and vegetated treatment system (VTS) consisting of a vegetated stream reach, a constructed wetland and a retention pond inside a flood detention basin. Shading is based on a digital terrain model with a resolution of 1 m². Location of drainage pipes is according to Gassmann et al. (2012).

## 2.2    Target compounds

In this study, we focused on 6 target compounds including the fungicides boscalid and penconazole, the herbicides metazachlor and flufenacet, and the TPs metazachlor sulfonic acid (met-ESA) and metazachlor oxalic acid (met-OA). Selected physicochemical properties of the target compounds are listed in Table 1. According to the Pesticide Properties Data Base (Lewis et al., 2016) the contaminants can be classified as low (boscalid) to moderately soluble in water, very mobile (TPs) to slightly mobile (fungicides). The target fungicides are considered moderately fast degradable in the water phase and persistent in soils, while the target herbicides are considered stable in the water phase and non-persistent in soils. TPs of metazachlor are considerably more persistent in soil than their PC. The fungicides are considered stable with respect to hydrolysis but degradable via photolysis, while the herbicides are stable regarding both.

Table 1: Physicochemical properties of the target compounds according to the Pesticide Properties Data Base (Lewis et al., 2016) including chemical formular, water solubility, organic carbon sorption coefficient ($K_{OC}$) as well as half lives in water ($T_{50}$ water) and soil ($T_{50}$ soil).

| | *Fungicides* | | *Herbicides* | | *TPs* | |
|---|---|---|---|---|---|---|
| | **Boscalid** | **Penconazole** | **Metazachlor** | **Flufenacet** | **met-ESA** | **met-OA** |
| **Chemical formula** | $C_{18}H_{12}Cl_2N_2O$ | $C_{13}H_{15}Cl_2N_3$ | $C_{14}H_{16}ClN_3O$ | $C_{14}H_{13}F_4N_3O_2S$ | $C_{14}H_{17}N_3SO_4$ | $C_{14}H_{15}N_3O_3$ |
| **Molecular mass (g mol$^{-1}$)** | 343.2 | 284.2 | 277.8 | 363.3 | 323.4 | 273.3 |
| **Solubility  (mg l$^{-1}$)** | 4.6 | 73.0 | 450.0 | 51.0 | - | - |
| **$K_{OC}$ (ml g$^{-1}$)** | 772.0 | 2205.0 | 79.6 | 273.3 | 5.0 | 24.6 |
| **$T_{50}$ Photolysis (d)** | 30.0 | 4.0 | *stable* | *stable* | - | - |
| **$T_{50}$ Hydrolysis (d)** | *stable* | *stable* | *stable* | *stable* | - | - |

| | | | | | | |
|---|---|---|---|---|---|---|
| $T_{50}$ Water (d) | 5.0 | 2.0 | 216.0 | 54.0 | - | - |
| $T_{50}$ Soil (d) | 246.0 | 117.0 | 10.8 | 19.7 | 123.3 | 90.0 |

## 2.3 Discharge measurement and sampling procedure

Stream flow was measured every minute between April 2016 and September 2017 at two gauges about 200 m upstream of the treatment system (G1) and at its outlet (G2). Water levels at G1 were recorded inside a 1.37 m standard H-flume (Bos, 1989) by means of a pressure transducer (Decagon CTD-10) and related to discharge using a standard rating curve. At G2 water levels were measured in a rectangular cross-section 2 meters ahead of the detention basin outlet by a radar gauge (Vegapuls 61). The corresponding rating curve for G2 accounted for complete submergence of the control gate valve (Peter, 2005). Pesticide monitoring at G1 and G2 consisted of 5 manual sample collections during stationary flow conditions and 10 automated event samplings during discharge events. Event sampling was triggered when a water level increase of more than 3 cm/h was registered at G1. An automatic sampler (ISCO 3700) started to fill pairs of 900 ml glass bottles at 0, 0.5, 1, 2, 6, and 12 hours after activation. A second automatic sampler (ISCO 3700) was launched at G2 following the same sampling scheme but with a time lag of one hour to account for transit between G1 and G2. All samples were recovered from the study site within 24 h after sampling and cooled until analysis. Sampling was complete except for one case. Due to accident we lost the first sample of event 9 (2017/10/08 03:30). As concentrations in the first samples were usually very low (Figure 3) and not considered to markedly influence mass calculation, we assumed that all contaminants in this sample had zero concentration and left this event in our data set.

## 2.4 Analytical methods

The following analytical methods were used for determining pesticide levels in the water samples. Analytical standards of boscalid (99.9%), penconazole (99.1%), metazachlor (99.6%), and flufenacet (99.5%) and the internal standards Diuron-D6 (99 %) and Terbutryn-D5 (98.5 %) already dissolved in acetonitrile (100 µg mL-1) were purchased from Sigma-Aldrich Chemie GmbH (Steinheim, Germany). Met-ESA (95 %) and met-OA (98.8 %) were received from Neochema (Bodenheim, Germany). Acetonitrile (LC-MS grade; VWR International GmbH, Darmstadt, Germany) was used as organic mobile phase in chromatography and for the preparation of stock solutions. Aqueous mobile phase was prepared with ultrapure water (Membra Pure, Germany; Q1:16.6 mΩ and Q2: 18.2 mΩ.

Preparation of environmental samples (approx. 1 liter) was done by filtering with a folded filter (type 113 P Cellulose ø 240 mm). Supernatant was spiked with the internal standard Diuron-D6 (10 µl of 10 mg L-1). Extraction procedure was a solid phase extraction (SPE). Cartridges (CHROMABOND® HR-X 6 mL/200 mg) were conditioned with 10 mL methanol and washed with 10 mL pure water. 90 µL of the extract were spiked with 10 µl of Terbutryn-D5 as an internal standard. Each sample was a double determination. Measurements of environmental samples were conducted with a Triple Quadrupole (Agilent Technologies, 1200 Infinity LC-System and 6430 Triple Quad, Waldbronn, Germany). Mobile phases were 0.01%

formic acid (A) and acetonitrile (B) with a flow of 0.4 mL min⁻¹. Gradient was as follows: 0-1 min (10% B), 1-11 min (10-
50% B), 11-18 min (50-85% B), 18-21 min (85-90% B), 21-24 min (90% B), 24-26 min (90-10% B) and 26-30 (10% B). A
NUCLEODUR® RP-C18 (125/2; 100-3 μm C18 ec) column (Macherey Nagel, Düren, Germany) was used as stationary phase
with a set oven temperature of T = 30°C. Calibration curve were prepared in pure water. The linearity was evaluated by
preparing three curves with ten calibration points in the range 1 - 500 µg/L. The standard curves were then extracted according
to the protocol and analyzed using LC-MS/MS. The calculated linear regression values ($R^2$) were very good with $R^2$-values >
0.999. The linearity between peak area and concentration of substances were obtained in a range of 0 - 5 µg L⁻¹. Hence limits
of detection (LOD) and quantitation (LOQ) were calculated with DINTEST (2003) according to DIN 32645 considering an
enrichment factor of 5000. LOD and LOQ amounted to 0.4 and 1.3 ng L⁻¹ (boscalid), 0.3 and 0.9 ng L⁻¹ (penconazole), 0.3 and
1.2 ng L⁻¹ (metazachlor), 0.4 and 1.3 ng L⁻¹ (flufenacet) as well as 0.6 and 2.2 ng L⁻¹ (met-ESA) and 0.5 and 1.6 ng L⁻¹ (met-
OA) considering an enrichment factor of 5000. A detailed analysis of measurement precision can be found in the supplementary
material (Text S1).
**2.5      Data analysis and calculations**
2.5.1      Identification of patterns in input concentrationIdentification of patterns in input chemographs was done by k-medoids
cluster analysis - a variation of the commonly applied k-means algorithm. Both approaches partition the elements of a dataset
into a predefined number k of clusters by attributing the elements to the cluster with the nearest cluster center. Optimal
clustering is achieved by iteratively updating cluster centers and minimizing distance between data points and cluster centers.
K-medoids differs from k-means as it uses existing points (medoids) as cluster centers instead of means and is considered more
robust against extreme values and outliers (Han et al., 2012). A total of 58 concentration sequences was included in the analysis,
consisting of 10 sequences per target compound, except for flufenacet which did not exceed LOQ in two events. Prior to cluster
analysis, data was normalized by the maximum of each chemograph to promote that clustering represented shape, rather than
differences in absolute concentration. The analysis was done using the software R (R Core Team, 2019) (version 3.6.1) using
the 'pam' (partitioning around medoids) function from the 'cluster'-package (version 2.1.0) (Maechler et al., 2019). We tested
clustering for k ranging between 2 and 10, the final number was determined by both visual inspection of the clusters and
assessment of explanatory benefit per additional cluster (elbow method). As a result we found that k=4 resulted in the best
partition.
**2.5.2      Contaminant mitigation**
Contaminant retention was assessed in terms of both peak-concentration reduction rate ($R_C$) and mass removal rate ($R_M$). $R_C$
was calculated in accordance with other studies (Eq. 2), e.g. Elsaesser et al. (2011), Stehle et al. (2011) and Passeport et al.
166 (2013):

$$R_C = \frac{C_{in,max} - C_{out,max}}{C_{in,max}} \cdot 100 \% \, ,$$

( 1 )

where $C_{in,max}$ and $C_{out,max}$ are peak concentrations registered at the inlet and outlet sampling points, respectively. $R_M$ was
calculated analogously from the input ($M_{in}$) and output contaminant mass ($M_{out}$):

$$R_M = \frac{M_{in} - M_{out}}{M_{in}} \cdot 100\ \% ,$$

(2)

Contaminant masses were calculated from discharge at G1 and G2 and linearly interpolated contaminant concentrations. As
water level data from G2 showed evidence for inaccuracy during low flows as result of the rectangular shape of the measuring
cross-section at G2, we did not assess mass removal during stationary flow conditions. As we did not sample the wetland
sediments or plants, the mass removal rate calculated following the above procedure describes the relative difference of
dissolved contaminant mass entering and leaving the wetland within the duration of the sampling procedure. It is therefore not
independent of the wetland's water balance:

$$W_B = \frac{Q_{in,mean} - Q_{out,mean}}{Q_{in,mean}} \cdot 100\ \% ,$$

(3)

where $Q_{in,mean}$ and $Q_{out,mean}$ are the discharge at G1 and G2, respectively, averaged over the duration of the sampling procedure
at both gauges. $W_B$ was positive, if more water entered the wetland than left the wetland during the sampling procedure, and
negative in the opposite case.
**2.5.3    Dispersion sensitivity of chemographs**
We defined a dispersion sensitivity index as follows:

$$i_{DS} = \frac{C_{in,max} - C_{in,n}}{C_{in,max}} ,$$

(4)

where $C_{in,n}$ is the concentration in the last sample and $C_{in,max}$ is the peak concentration of a chemograph recorded at the inlet of
the VTS (G1). In other words, $i_{DS}$ represents the fraction of the concentration peak that can potentially be flattened by
dispersion.
**3    Results**
**3.1    Contaminant mobilization**
Contaminant concentrations in stream water (G1) differed clearly depending on the flow conditions (Figure 2). During
stationary flow, concentrations of boscalid and the TPs of metazachlor ranged in the order of tens of nanograms, while
penconazole, metazachlor and flufenacet only occasionally exceeded the LQ. During discharge events in contrast, peak
concentrations varied from a few nanograms (flufenacet) to several milligrams per liter (boscalid) spanning a range of 6 orders
of magnitude. Concentration increase during events compared to stationary flow was different among the compounds. Median
concentration of boscalid increased by a factor of 48, while concentrations of met-ESA and met-OA only increased by a factor
of 3 and 5, respectively. Similar patterns were found for contaminant mass. Contaminant mass mobilized in the catchment
during discharge events ranged from several hundreds of micrograms (flufenacet) to several hundreds of milligrams (boscalid)
and even several grams in exceptional cases (boscalid, metazachlor). Based on compound medians, about 76 times more
boscalid but only about 4 times more met-OA were transported during discharge events than during an equally long period
under stationary flow conditions.

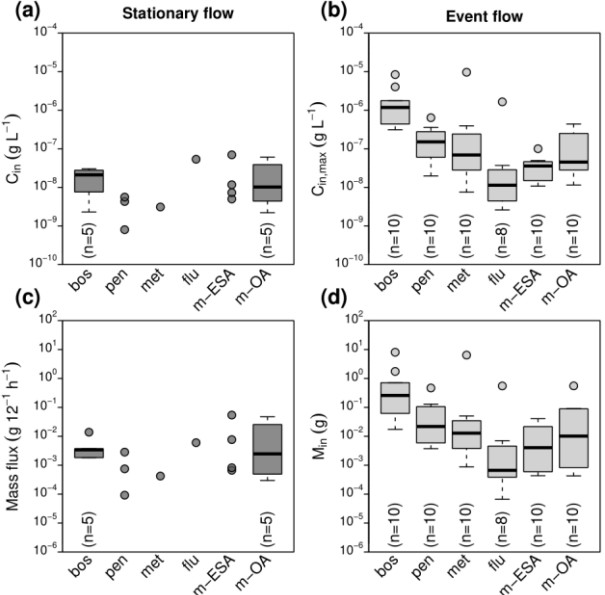


**Figure 2: Peak concentrations at G1 during stationary flow (a) and flow events (b) as well as contaminant mass flux during stationary**
**flow (c) and transported mass per event (d). Boxplots indicate median and interquartile range (IQR). Whiskers indicate extreme**
**points within 1.5 times the IQR from the boxes, circles indicate points outside this range.**
The 10 events were characterized by different discharge magnitudes and dynamics (Figure 3). Mean discharge during the
events ranged between 0.7 (E10) and 32.0 L s$^{-1}$ (E2) with respective peak values between 4.4 (E10) and 199.7 L s$^{-1}$ (E2). The
recorded event hydrographs included events with one single discharge peak (E4, E5, E6, E10), with one major peak followed
by one or more secondary peaks (E2, E3, E7, E9), and events in which a major peak followed an earlier smaller peak (E1, E8).
In most cases discharge had recessed to pre-event levels by the end of the 12-hour sampling procedure, only E1 and E2 showed
ongoing flow recession. In many cases, concentrations in the final event samples were still elevated compared to pre-event
conditions. However, due to flow recession, mass flux was usually very low by the time the last sample was collected
(Figure S1).

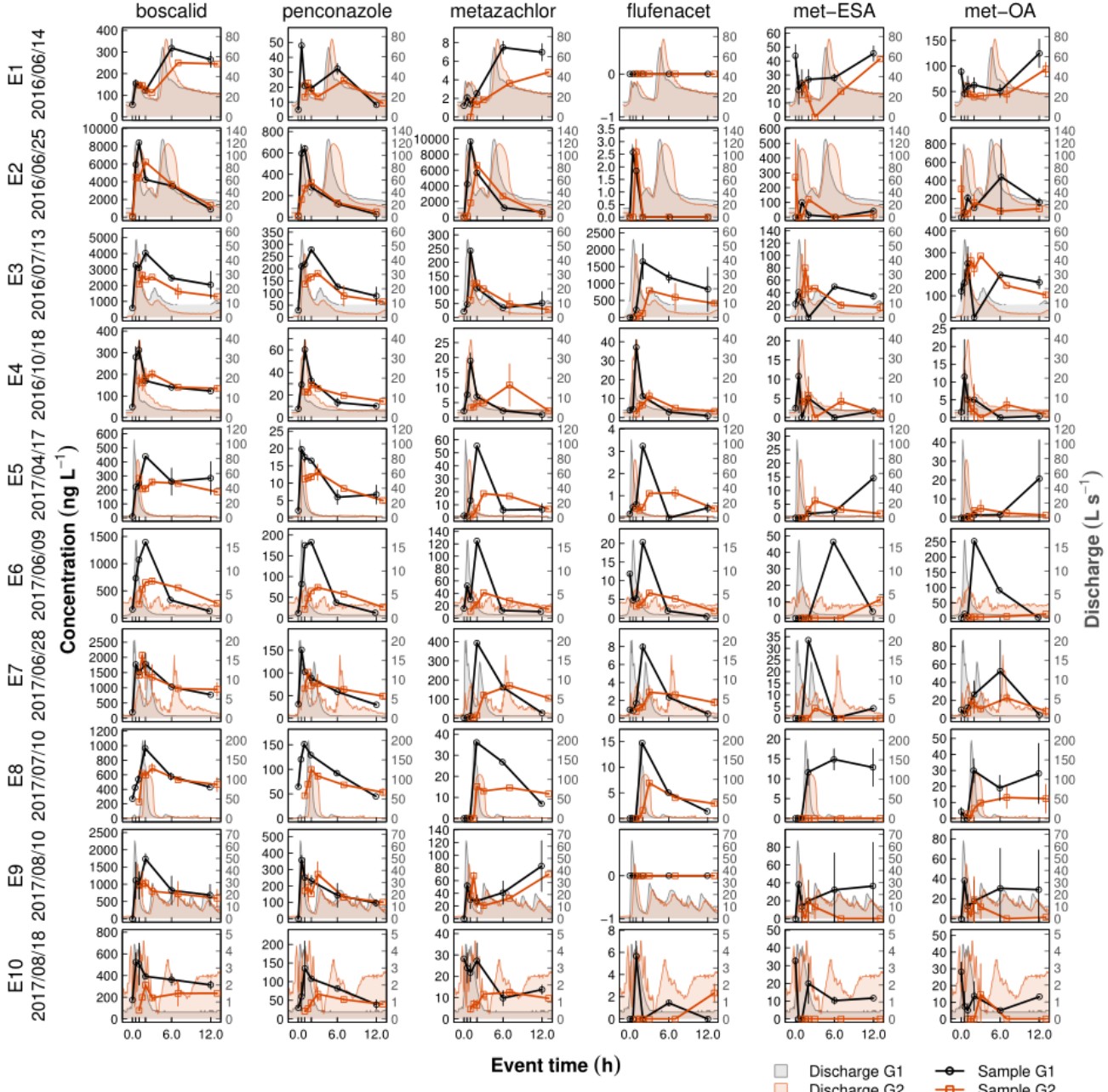

**Figure 3: Contaminant concentration at the inlet gauge G1 and the outlet gauge G2 of the 6 target compounds during 10 discharge**
**events. Data points represent means of duplicate samples and standard deviation (error bars).**

## 3.2   Patterns in chemographs

Cluster A (Figure 4) was characterized by absence of a clear peak during the first two hours of sampling but elevated concentrations during later times, resulting in low $i_{DS}$. Cluster B showed a quick response, i.e. concentrations increased sharply within the first 30 minutes. Concentrations were the highest of all clusters and still elevated in the last sample compared to pre-event levels. Cluster C was characterized by a clear peak within the first two hours and a low tailing and was the cluster with highest median $i_{DS}$. Cluster D showed the most inconsistent pattern and maximum concentrations appeared later compared to clusters B and C. A relatively clear pattern was evident in the attribution of compounds to the clusters. Chemographs of the fungicides boscalid and penconalzole were mainly assigned to cluster B, while the herbicides and the TPs were assigned to the remaining three clusters. Cluster A was composed of herbicide and TP chemographs, particularly from events with multiple discharge peaks. Cluster D represented chemographs of herbicides and TPs mainly during the events E5 to E8 which were all characterized by sharp discharge peaks during periods of generally low flow (Figure 3). Almost all chemographs of the events E2 and E4 were attributed to cluster C.

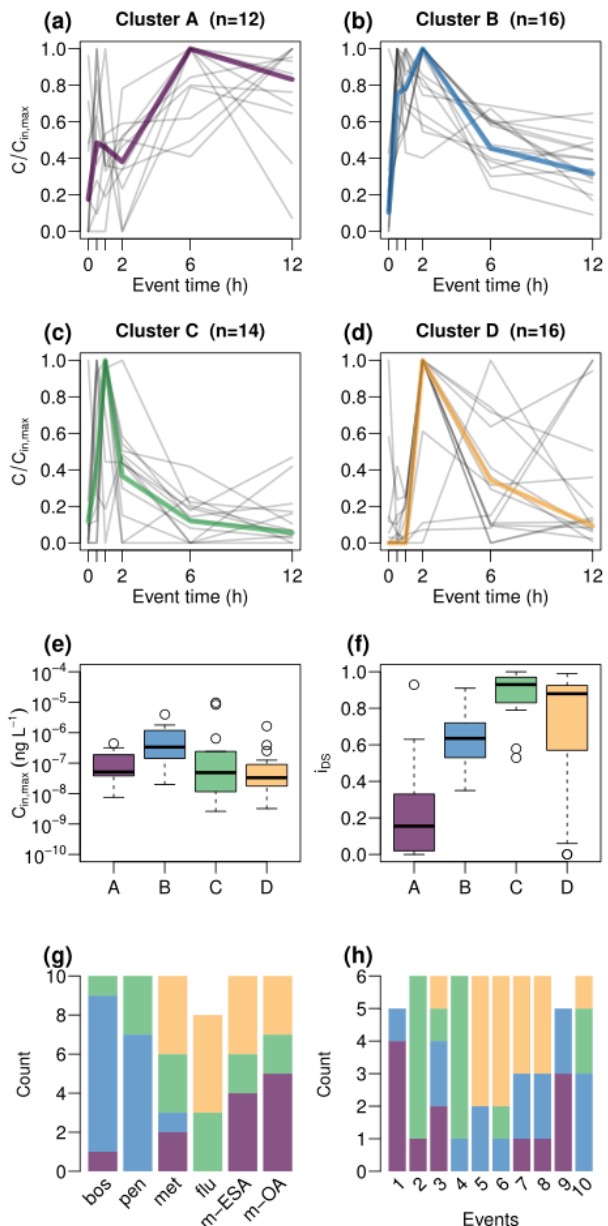

224

**Figure 4: Clustered event chemographs as well as maximum input concentration, dispersion sensitivity index, and attribution of compounds and events to the different clusters**

### 3.3 Contaminant mitigation in the wetland

Contaminants were mitigated in the wetland in terms of both $R_C$ (Eq.1) and $R_M$ (Eq.2). $R_C$ (Figure 5a) was close to zero for boscalid and poorly constrained for the remaining compounds during stationary conditions, partly due to insufficient number of detections. During discharge events, in contrast, peak concentrations of all compounds were clearly reduced. While $R_C$ was narrowly constrained for the fungicides and herbicides, TPs exhibited higher variability. Mean $R_C$ and corresponding standard

deviations were 29.8 ± 18.4 % (boscalid), 42.1 ± 11.5 % (penconazole), 47.9 ± 16.4 % (metazachlor), 53.8 ± 22.6 %
(flufenacet), 29.5 ± 84.7 % (met-ESA), and 47.9 ± 29.5 % (met-OA), respectively. $R_C$ was clearly different among chemograph
clusters with lowest values in cluster A and highest values in cluster D (Figure 5b). Moreover, $R_C$ was higher for the lower
half of peak concentrations than for the upper half (Figure 5c) and systematically increased with dispersion sensitivity (Figure
5d). $R_C$ was also related to discharge conditions. Highest $R_C$ values were reached, when mean discharge was low (Figure 5e)
but the ratio of maximum to mean discharge was elevated (Figure 5f), i.e. in events characterized by low pre-event discharge
and sharp discharge peaks (in particular events attributed to cluster D in Figure 4). Although there was evidence for major
water surpluses and deficits in the event water balance between G1 and G2, particularly in events with low discharge such as
events E6 and E10, an imbalanced water balance had only minor effects on $R_C$ (Figure 5g). We did not find clear relationships
between $R_C$ and compound properties such as $Kf_{OC}$, water solubility or soil half-live (Figure 5h-j).
Relative mass removal during discharge events (Figure 6a) resulted in smaller rates and higher variability compared to $R_C$.
Mean $R_M$ and corresponding standard deviations were 7.7 ± 29.6 % (boscalid), 17.3 ± 26.0 % (penconazole), 18.1 ± 27.8 %
(metazachlor), 27.0 ± 28.1 % (flufenacet), 35.2 ± 68.4 % (met-ESA), and 44.0 ± 28.7 % (met-OA). These values show that
mass removal was limited for most compounds. Although the general pattern in $R_M$ for the different compounds was similar
to $R_C$, behavior of $R_M$ among the chemograph clusters was different. While $R_C$ increased from cluster A to cluster B and C,
$R_M$ decreased (Figure 6b). Cluster D exhibited high values of both $R_C$ and $R_M$. No clear response was found to different levels
of input mass (Figure 6c), however, median $R_M$ was lowest when peaks in chemographs were sharpest (Figure 6d). This means
the relationship of $R_M$ to increasing sharpness of chemograph peaks was inverted compared to $R_C$. $R_M$ was not obviously
related to discharge dynamics, neither to mean discharge (Figure 6e), nor to the ratio of maximum to mean discharge (Figure
6f). Disregarding events with very low discharge (events E6 and E10), it seemed possible that much of $R_M$ was the result of
water imbalances during the events (Figure 6g). However, $R_M$ of most chemographs plotted above the 1:1 line of $R_M$ and
relative water balance, indicating that $R_M$ was higher than water imbalance would explain. $R_M$ showed a tendency to decrease
with increasing $Kf_{OC}$ (Figure 6h), but no clear pattern was found for solubility (Figure 6i) and soil half-live (Figure 6j).

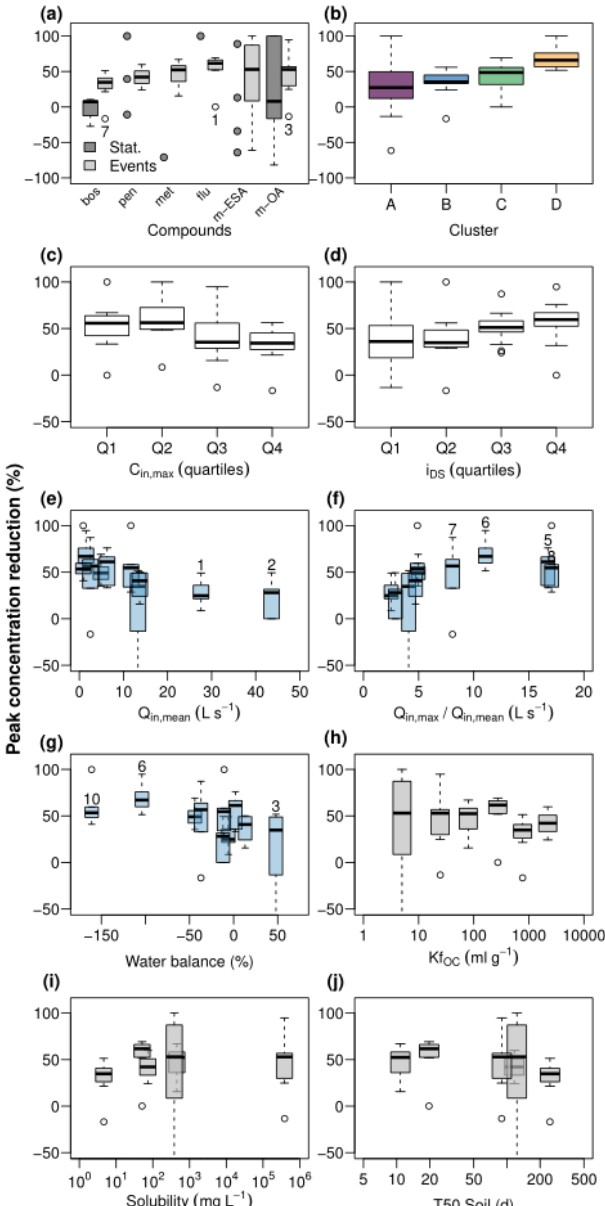


Figure 5 Contaminant peak concentration reduction in the wetland (a) during stationary and event flow conditions and its relationship to chemograph properties, discharge conditions, and physiochemical properties of the target compounds. Chemograph properties include clustering (b), peak concentrations (c), and ratio of concentrations during the peak and in the tailing (d). Discharge conditions include mean discharge at G1 (e), ratio of maximum to mean discharge at G1 (f), and water balance between G1 and G2 (g). Event numbers are shown for selected events. Compound properties include organic carbon sorption coefficient (h), solubility in water (i) and soil half-live (j).

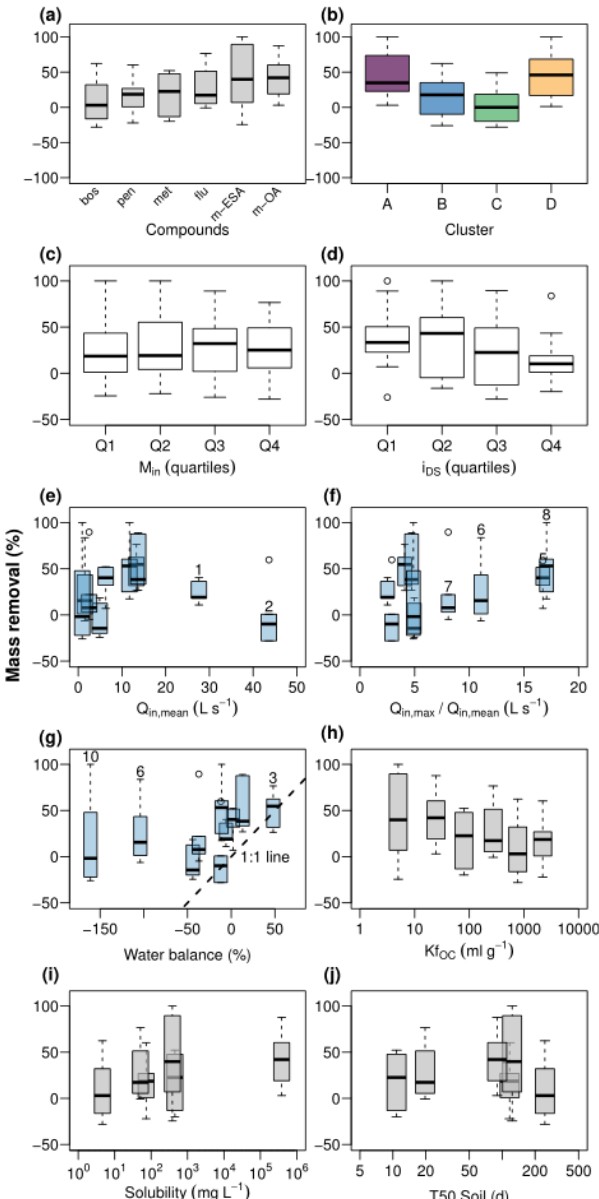

262

**Figure 6: Contaminant mass removal in the wetland (a) during event flow conditions and its relationship to chemograph properties, discharge conditions, and physiochemical properties of the target compounds. Chemograph properties include clustering (b), peak concentrations (c), and ratio of concentrations during the peak and in the tailing (d). Discharge conditions include mean discharge at G1 (e), ratio of maximum to mean discharge at G1 (f), and water balance between G1 and G2 (g). Event numbers are shown for selected events. Compound properties include organic carbon sorption coefficient (h), solubility in water (i) and soil half-live (j).**

## 4 Discussion

### 4.1 Monitoring setup and associated uncertainties

Regarding chemographs and calculation of $R_C$, uncertainties arose from timing and frequency of sampling and analytical error, and additionally from discharge measurement when calculating masses and $R_M$. Analytical methods used in this study usually produced very consistent results so that variability in concentrations of parent compounds in duplicate samples was low (sd < 10 %). However, in individual samples collected at G1 analytical variability was elevated for met-ESA and met-OA (Figure 3), reducing confidence in concentrations and the derived measures $R_C$ and $R_M$ of TPs in the affected chemographs (E2, E5, E7, E8, E9). Uncertainty related to timing and frequency of sampling can hardly be quantified but certainly depends on how well the sampling intervals captured variability in concentrations during flood events and how well the time lag between upstream and downstream sampling matched the residence time of solutes in the wetland. Lefrancq et al. (2017) assessed the effect of sampling frequency in pesticide monitoring data collected during runoff from a single vineyard and found that acute toxicity of pesticide flushes was underestimated up to 4-times when calculated from event means and up to 30-times when calculated from random samples. Although these data were collected on the plot scale and we assume that variability in our catchment is lower due to longer flow paths and mixing processes on the catchment scale, uncertainty of the chemographs in our study could have been reduced by increasing sampling frequency. Regarding the timing of upstream and downstream sampling, there is evidence that water residence time in the wetland was in fact shorter than one hour. The observation that for quickly responding compounds, such as boscalid, concentration in the first sample at G2 was often elevated compared to the first sample at G1 indicates that the contaminant flush hat already reached G2 when sampling started. This did not influence determination of $C_{out,max}$ and $R_C$ in the outlet of the wetland, as concentrations were still rising from the first to the second sample (Figure 3). However, effects on $M_{out}$ were higher, since a relevant fraction of contaminant mass leaving the wetland was not registered and thereby caused overestimation of $R_M$ (Figure S1). Another source of uncertainty exclusively affecting contaminant mass and not concentrations was the use of different gauging systems at G1 and G2. Different shapes of the measurement cross-section (triangular at G1 and rectangular at G2) caused G2 to be less precise and water imbalances on the event scale, particularly when flow was low. Summarizing the setup constraints above, we have high confidence that the experimental setup produced realistic chemograph shapes and captured peak concentration reasonably well, but are less confident regarding contaminant loads.

### 4.2 Mobilization of contaminants and formation of distinct chemographs

Peak concentrations of mobilized contaminant flushes were different depending on the compound. This may be due to the application of different amounts and due to temporal patterns of application. The fact that maximum concentrations of metazachlor and flufenacet in specific events exceeded concentrations during most other events by a factor of more than 100, suggested application of these compounds shortly before the onset of runoff. Despite highly variable application patterns, our cluster analysis resulted in four groups with similar chemograph shape. Many factors have been shown in literature to influence the mobilization of pesticides in catchments, including catchment properties, event properties and physiochemical compound

properties. As catchment properties we here consider factors associated with runoff generation such as catchment geometry, terrain slopes, and in particular the delineation of areas where different compounds were applied. The interplay of these factors defines hydrological activity and connectivity (i.e. by shortcuts like roads and drainage pipes) of critical source areas for different compounds (Doppler et al., 2012; Gomides Freitas et al., 2008). Event properties include intensity and dynamics of rainfall (Imfeld et al., 2020) and subsequent runoff (Doppler et al., 2014). Relevant physiochemical compound properties are e.g. mobility and degradability (Gassmann et al., 2015).

These properties are reflected to varying degrees in the results of the cluster analysis. Cluster A was characterized by a quick response and a concentration plateau towards the end of sampling and was mainly composed of TPs. The fact that concentration maxima in cluster A were delayed compared to fungicides (cluster B), although their parent compounds were applied closer to the stream in the flat valley bottoms, suggests that they were transported with a slower flow components. Due to flatter terrain, surface runoff played a less important role and the main transport pathway was subsurface flow. Where fields were undrained, however, transit time of water from the infiltration point to the stream would likely exceed the temporal scale of event sampling. Most of the water reaching the stream from the fields in the valley during discharge events would therefore be pre-event water, enriched in TPs formed in the soil, corresponding to the formation site of TPs of the chloracetamide herbicides to which metazachlor belongs (Mersie et al., 2004). Seepage of pre-event TP-rich water thus explains the immediate response of chemographs in cluster A. The quick response was often followed by a local concentration minimum between samples 2 and 5, i.e. between 30 min and 6 h after sampling was initialized. Coincidence of this minimum with concentration peaks of fungicides might suggest dilution of TP concentration by mixing with event water carrying high loads of fungicides but less TPs of metazachlor.

Cluster B represented differences between fungicides and the remaining compounds. Considering land use distribution in the studied catchment, it is unclear whether this partition reflects different compound properties or catchment properties or both. The fact that concentration in cluster B quickly increased with discharge (within 30 minutes) is in line with fast transport from the vineyard terraces to the stream via roads and drainage pipes as described by Gassmann et al. (2012) for suspended solids in the studied catchment. Along such preferential pathways, compound properties, such as sorption affinity, may be less important (Gomides Freitas et al., 2008) compared to e.g. percolation through the soil with intense contact to sorption sites in the soil matrix. Moreover, fungicides are applied by sprayers into the foliage and can drift to e.g. paved surfaces from which they can be quickly mobilized by subsequent rainfall (Lefrancq et al., 2013). We therefore hypothesize that cluster B was mainly produced by surface flushing and fast transport pathways of fungicides. This explained the quick rise and subsequent decline in concentrations (concurrent with plateaus produced by slower flow components in cluster A).

Cluster C was composed of chemographs of all compounds but mainly from events E2 and E4 indicating event dependence. Two aspects were found to support this idea. First, there was a secondary discharge peak in event 2 that did not contribute much in terms of contaminant concentration but rather caused dilution and produced particularly flat chemograph tails. Second, peaks of herbicides and TPs were less delayed compared to fungicides. This may be the result of recent herbicide application and active surface runoff in the flat valleys. Timing of pesticide application was identified as the main export driver of currently

used pesticides by Imfeld et al. (2020) who performed a cluster analysis on rainfall data from a headwater vineyard catchment.
Based on the magnitude of discharge and amount of mobilized contaminants (concentration of metazachlor $\approx$ 10 µg L$^{-1}$), both
explanations seem plausible in event E2. Event E4, however, did not show particularly high herbicide concentration nor a
secondary discharge peak. Although it is obvious that chemograph shapes in cluster C differed from the other clusters,
unfortunately, the responsible factors remain unclear.
Cluster D included chemographs of both herbicides and TPs and presented a clear peak that was often defined by a single
sample 2 h after the beginning of the event. In contrast to cluster A, cluster D was characterized by a single sharp discharge
peak (except in event E7 where a second peak occurred shortly after the first) and mainly included chemographs during periods
of low flow. Our interpretation is that cluster D represented flow events in which no dilution of herbicide and TP fluxes by
fungicide fluxes or secondary discharge peaks occurred. Low pre-event discharge in cluster D compared to cluster A may
indicate low water levels which may have caused a slower response as no enriched pre-event water was released from the soils
in the valleys.
The unclear interpretation of cluster C suggest that we missed important factors for the formation of chemographs. In fact,
variables like spatial distribution of rainfall or pesticide application rates and timing (Imfeld et al., 2020) and possibly other
factors likely influenced chemograph shapes. Knowing all these variables would not change the results produced by the
clustering algorithm but rather increase our ability to interpret them. Nevertheless, the cluster analysis helped to explore how
the catchment and processes therein influenced concentration signals of mobilized contaminants. Particularly, the analysis
helped to understand under what conditions and for which pollutant sharp-peaked chemographs, associated with high acute
toxicity, can be expected. We therefore see a high potential of this type of analysis for the identification of influential factors
for contaminant mobilization in other catchments, although these factors may not be universal but catchment-dependent.

### 355   4.3      Mitigation efficiency and chemograph shape

#### 356   4.3.1      Peak concentration reduction

We hypothesized that peak concentration reduction in the VTS will be highest for chemographs with the sharpest peaks, i.e.
for the chemographs that were most sensitive to dispersion. And indeed we found a systematic relationship between $R_C$ and
both $i_{DS}$ and chemograph clusters. Although the relationship of clusters and $R_C$ largely reflected the relationship between $R_C$
and $i_{DS}$, it is surprising that $R_C$ was clearly highest in cluster D and not in cluster C which presented better defined peaks and
slightly higher $i_{DS}$ per cluster (Figure 4f). Critical inspection of input chemographs shows that in several chemographs of TPs
(met-ESA and met-OA in event E4 and met-OA in event E8) elevated concentrations in the last samples exhibited high
analytical errors and did not appear in the outlet chemograph. These dubious samples caused low $i_{DS}$ but substantial $R_C$ and
thus contributed to variability in $i_{DS}$ despite high values of $R_C$ in cluster D. We therefore do not consider the deviation from
the expected cluster ordering contradictory but to result from increased uncertainty in cluster D as mentioned earlier. In
contrast, the hypothesized relationship between RC and chemograph shape was demonstrated for both $i_{DS}$ and chemograph
clusters, the latter of which also integrates shape aspects that go beyond $i_{DS}$, e.g. timing of peaks. Overall, the values of $R_C$
found in our study compare with field data from vegetated buffers (Bundschuh et al., 2016; Stehle et al., 2011) and are in the
range of those found in vegetated stream mesocosms by Elsaesser et al. (2011) and Stang et al. (2014) who both attributed
most of the observed peak reduction to dispersion.
In addition, we found relationships between and $R_C$ and discharge dynamics, i.e. $Q_{mea}$ and ratio of $Q_{max}$ to $Q_{mean}$. The influence
of discharge on $R_C$ may be two-fold. First, increasing flow reduced residence time and hydraulic efficiency, i.e. short circuiting
reduced the potential for dispersion and interaction with wetland sediments or plants. Second, the fact that chemographs of
events with high $Q_{max}$ to $Q_{mean}$ ratios were attributed to cluster D suggests that discharge dynamics influenced the shape of the
chemograph at the wetland inlet. This means, the influence of discharge may also be indirect by promoting the formation of
sharp-peaked chemographs with high potential for peak reduction.
In contrast to other studies, we did not find clear relationships of $R_C$ to and physiochemical properties of compounds such as
sorption affinity (Stehle et al., 2011; Vymazal and Březinová, 2015) or solubility (Bundschuh et al., 2016). The absence of
such relationships may partially be due to the low number of different target compounds in our study (n=6). However, given
the short time lag between sampling at the inlet and outlet of the wetland ($\Delta t = 1h$), it seems logical that no relevant sorption
or degradation occurred within this period. For comparison, in batch experiments by Gaullier et al. (2018) adsorption
equilibrium for boscalid (compound with second highest $Kf_{OC}$ in our study) was only reached after 24 h. Despite the relatively
narrowly confined $R_C$ values of the parent compounds, we do not consider physiochemical compound properties as major
drivers of $R_C$ in our VTS.

### 4.3.2 Contaminant mass removal

For $R_M$ we found a different pattern among the chemograph clusters than for $R_C$. $R_M$ was apparently higher in clusters A and
D than in clusters B and C. However, the clusters indicating substantial mass removal were those with increased uncertainty
regarding compound mass. Cluster A often showed relevant mass flux at the end of sampling (and presumably beyond) which
we did not account for. Cluster D contained dubious data points of TPs and poorly defined peaks outside the periods of high
sampling frequency. In addition, due to overestimation of solute travel time in the wetland in the monitoring setup, the rising
limp of the mass flux signal at G2 was often not adequately captured by the sampling scheme, causing underestimation of
downstream event mass and overestimation of mass loss. In absence of any clear relationship with compound properties,
discharge dynamics or chemograph shape, this suggests that the assessment of contaminant masses was subject to systematical
errors and that the apparent mass loss found in our study should therefore not be over-interpreted.
In earlier studies, Lange et al. (2011) and Schuetz et al. (2012) observed a 15-30 % mass loss of the fluorescent tracer
sulforhodamine-B in the wetland subsection of the current VTS. These results indicate a general potential for sorption of
organic compounds in this system, but represent an earlier succession state of the wetland and stationary flow conditions with
much longer residence times. Also in the current VTS kinetic sorption of contaminants may have occurred but sorption
equilibrium was certainly not reached (Gaullier et al., 2018). Thus the effect of sorption did not reach its full potential. In fact,
other studies reported limited mass removal in wetlands with comparable residence times. Ramos et al. (2019) did not find
relevant $R_M$ in two surface flow wetlands with residence times between 45 min and 6 h in England. In contrast, Passeport et
al. (2013) found $R_M$ between 45 % and 96 % in a constructed wetland with a residence time of 66.5 h. However, their
contaminant mass loss coincided with loss of water (45 %). Mesocosm experiments by Elsaesser et al. (2011) and Stang et al.
(2014) showed strong concentration reduction but only very limited and temporary mass removal at residence times of a few
hours. In summary, these findings suggest that the potential for mass removal in wetland systems like the one studied here is
rather limited. However, wetlands have been shown to reduce contaminant mass, when residence times are sufficiently long
(Gregoire et al., 2009) or when operated in batch mode (Tournebize et al., 2017; Moore et al., 2000; Maillard et al., 2016).
**4.4    Conclusions**
In agreement with other studies this investigation shows that VTSs with short water residence times of up to several hours may
cause substantial reduction of peak concentrations of contaminants mobilized during discharge events. This implies an efficient
reduction of acute toxicity for receiving aquatic ecosystems. In the present VTS the reduction of concentration peaks was
mainly controlled by dispersion and was more pronounced for sharp-peaked than for flat input chemographs. In contrast,
contaminant mass loss was rather limited, mainly due to the fact that short residence times did not allow for considerable
sorption or transformation. Clustering of chemographs revealed that chemograph shapes were associated with source areas,
input pathways and discharge dynamics. This highlighted the role of chemographs as links between processes in catchments
and in receiving aquatic systems. The presented cluster analysis helped to understand why and for which pollutant sharp-
peaked chemographs could be expected. Such sharp-peaked chemographs produce high acute toxicity in aquatic ecosystems
but at the same time can efficiently be mitigated in VTSs. While the factors controlling chemograph shape may be different in
different catchments, the effect dispersion exerts on these signals is universal.
**Data availability**
Contaminant data used in this study is available as supplementary material.
**Author contributions**
JL planned the monitoring concept. JG performed the data analysis and prepared most of the manuscript in cooperation with
JL. . OO and KK facilitated the sample analysis and OO wrote the section on analytical methodology.
**Competing interests**
The authors declare that they have no conflict of interest.

**Financial support**

This research has been supported by the Federal Ministry of Education and Research (BMBF) (grantno. 02WRM1366B) and the Water Network Baden-Württemberg funded by the Ministry of Science, Research and Art of the State of Baden-Württemberg. The article processing charge was funded by the Baden-Wuerttemberg Ministry of Science, Research and Art and the University of Freiburg in the funding program Open Access Publishing.

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
