# Peer review of "Pesticide peak concentration reduction in a small vegetated treatment system controlled by chemograph shape"

_Hydrology and Earth System Sciences, 2020_

## Referee Comment (RC1) · Anonymous Referee #1 · 23 Jul 2020

The authors present an interesting study design with almost experimental character. Based on Fig. 1, one can conclude that there is a spatial separation between different crops in the catchment (arable crops versus vineyards). Because these crops receive different pesticide treatments (with regard to timing, compounds) one gets a signal in the outlet of the catchment that is related to a certain spatial unit. This design has the potential for learning how spatial aspects, physico-chemical properties of pesticides and agricultural practices influence pesticide losses (dynamics, loss rates). This could also be relevant for better understand the functioning of the wetland as is the main purpose of this manuscript.

The manuscript analyses the effect of a wetland at the outlet of this agricultural catchment on pesticide transfer further downstream. Based on samples taken during 10 events and five baseflow periods, the retention capacity is studied for four active ingredients and two metabolites regarding the relative reduction of peak concentrations and pesticide mass transported downstream. Specifically, the authors investigated the effect of the shape of the chemograph at the inlet and a few compound properties and explain the differences in the chemograph shapes of different compounds by being applied to different parts of the catchment.

Monitoring pesticide dynamics in natural systems is demanding and requires substantial efforts in the lab and in the field. The observed pesticide behaviour in the environment is complex and often influenced by specific conditions at the local scale. Therefore it is valuable for science and practice if such observations get published from different locations and catchments because it broadens our understanding of the environmental fate of pesticides under different conditions.

Unfortunately, there a number of issues that limit the value of the manuscript in its current status. I describe my main concerns below and add some further details at the end.

**Scientific quality:**

The data analysis is rather superficial and several of important conclusions are not strongly backed up with data. This holds true for example for the interpretation of the different dynamic patterns identified (L. 210 - 221). For example, the authors hypothesise that fungicides from the upstream vineyards are more quickly transported to the stream than the herbicides applied closer to the creek in the valley bottom (L. 210 - 213). Why this should be the case remains unclear. Checking a previous publication on the catchments (Gassmann et al., 2012) reveals that there is a dense road network connected to the pipe drains. These structures are made explicit in the earlier publication but no linear structures are indicated in the catchment map of this manuscript. Given the fact that pesticide drift to non-target surfaces such as roads may be impor-
tant for pesticide transport in vineyards (Lefrancq et al., 2013) important aspects of the catchment are neglected and not included into the discussion of the results.

Also temporal aspects are only treated superficially. The shape of the chemograph of a pesticide is strongly influenced by the rainfall dynamics. Because the authors compare the dynamic of different pesticides across different events (this is not really evident from the main text, but see the SI), differences in concentration dynamics could also be strongly influenced by rainfall patterns and discharge behaviour. Unfortunately, no respective data is shown or discussed.

This holds also true for the timing between the last application of a pesticide and the rainfall event. The authors do not discuss this aspect and treat all compounds and all events the same (except for two events for flufenacet where too many samples < LOQ). However, inspection of the actual concentration data reveal strong differences in the concentration levels of the different compounds across all events. High concentrations of several hundred to thousands of ng L-1 were found for boscalid for all events, while flufenacet was only found in one event above 1600 ng L-1 but otherwise never above 40 ng L-1. Obviously, the history of the compounds since the last application was very different. Neglecting such aspects but interpreting the different relative concentration dynamics with respect to transport differences from different parts of the catchments is not very solid.

This also limits the value of the cluster analysis. The results in Fig. 3 show that a given compound appears in different clusters. However, this aspect is not properly discussed and no explanation is provided why this was the case nor what this implies for interpretation of these clusters. Additionally, it is not clear to which degree these results reflect the full spectrum of observed pesticide dynamics. The authors mention in the Method section (L. 190) that they have removed outliers based on purely statistical analyses. I don't think the procedure is sound (see below) and may bias the findings by excluding unusual - or simply rare - dynamics.

**HESSD**
Finally, the calculation retention rates raises a number of question marks, which may further impact the subsequent linear model for describing the retention efficiency of the wetland. First, given the measured concentrations (see SI, but also Fig. 3) it is evident that in many cases the last data point does not reflect baseflow concentrations after the event. Accordingly, the mass loss during the events may have been substantially larger in some cases. The extent of this effect depends on the unobserved concentration dynamics but also on the discharge. Unfortunately, no discharge data is provided that illustrate which part of the event hydrographs have actually be covered by the sampling. Second, the observed concentration levels demonstrate that for compounds such as flufenacet most events reflect conditions long after the last application. Accordingly, the concentration signal is rather weak and calculated retention is laced with considerable uncertainty. This aspect is not mentioned.

**Scientific significance:**

The issues listed above reduce the scientific significance of the manuscript. Additionally, there are questions about the scientific insight conveyed by the manuscript. There are two major issues:

**Relevance of the shape of the chemograph** The effect of the chemograph is almost a trivial finding. Given the short residence time in the wetland (about 1 h; see L. 81) degradation processes will be very limited and the main effect on peak concentrations in the outlet is expected by dispersion (as expected by the authors, see below), i.e. mixing water of different concentrations at the inlet. Obviously, the more variable the input, the larger the relative effect of mixing on the relative maximum concentrations. Actually, the authors have stated this outcome already in the Introduction as fact: "'*Peak concentration reduction will be stronger for a signal with pronounced peak and low background than for a signal with small peak and high background if both signals are exposed to the same dispersion.*" (L. 54 - 55). HESSD
This can also be easily shown with the following example: Let's assume that two compounds have the same background concentration ( $C_{min}$ ) but during a discharge event the concentration of compound B reaches higher values than compound A with the same relative dynamics during the event:

$$C^B = C_{min} + \alpha \left( C^A - C_{min} \right) \tag{1}$$

Assuming only mixing to occur, it is evident that the maximum outflow concentration ranges between  $C_{min}$  and  $C_{min} + \Delta C_{max}$  for compound A (indicated by the fraction  $\beta$  of the maximum possible value):

$$C_{out}^A = C_{min} + \beta \,\Delta C_{max} \tag{2}$$

$$C_{out}^B = C_{min} + \alpha \ \beta \ \Delta C_{max} \tag{3}$$

Accordingly, the relative peak concentrations in outflow compared to inflow is given as follows:

$$R^{A} = \frac{C_{min} + \beta \,\Delta C_{max}}{C_{min} + \Delta C_{max}} \tag{4}$$

$$R^{B} = \frac{C_{min} + \alpha \ \beta \ \Delta C_{max}}{C_{min} + \alpha \ \Delta C_{max}}$$
(5)

Taking the ratio of the relative peak concentrations shows that the relative concentration for the compound with the more pronounced peak is more strongly reduced:  $R^A > R^B$ :

$$\frac{R^A}{R^B} = \frac{C_{min} + \beta \,\Delta C_{max}}{C_{min} + \Delta C_{max}} \,\frac{C_{min} + \alpha \,\Delta C_{max}}{C_{min} + \alpha \,\beta \,\Delta C_{max}} = \tag{6}$$
(7)

 $= \frac{C_{min}^2 + (\alpha + \beta)C_{min}\Delta C_{max} + \alpha\beta(\Delta C_{max})^2}{C_{min}^2 + (1 + \alpha\beta)C_{min}\Delta C_{max} + \alpha\beta(\Delta C_{max})^2}$

This ratio is always  $\geq 1$  because  $\alpha + \beta \geq 1 + \alpha\beta$  (for  $\alpha \geq 1, 0 \leq \beta \geq 1$ )1.

In this context I am furthermore surprised that the authors did not use the model that was developed for the study wetland for evaluating the effect of different chemograph shapes (Schuetz et al., 2012). Although the wetland may have undergone changes since the last tracer experiments, it would be a useful null model for testing how different input signals influence the reduction in peak concentrations.

How to generalise the findings? It is difficult to generalise any of the findings reported in the manuscript because they are very context dependent and results such as the cluster analysis are very phenomenological. The aspect that the shape of the chemograph has an influence on the relative degree by which peak concentrations are reduced is quite evident for the type of system under investigation.

**Detailed comments:**

- L. 30: The phrase "'... which may be equally or more mobile, persistent and toxic than their PC ..." is misleading because it does not mention the general case that transformation products are less toxic.
- L. 93: How often were grab samples taken?
- L. 123: How adequate is it to only take one single isotope-labelled internal standard not corresponding to the target compounds? Checking these compounds in one of our current analytical methods, retention times vary between 16.7 (metazachlor-ESA) and 21.0 min (penconazole). Also the  $K_{OC}$  values vary by a factor of 400

HESSD

Interactive

comment

 $^{1}\text{with }\alpha=1+\delta\text{: }\alpha+\beta\geq1+\alpha\beta\Rightarrow1+\delta+\beta\geq1+\delta\beta+\beta$

between these two compounds. Please provide additional information supporting the assumption that terbutryn as an adequate internal standard (e.g., recoveries).

- L. 129 130: Please check the correct number of significant digits (can you measure with a precision of tens of picograms per litre?).
- L. 142: Please provide the version of R. I assume that you did not implement the algorithm but used kmeans () implemented in standard R.
- **L. 174 176:** Why did you include  $DT_{50}$  a priori? I'd recommend to leave it in. The subsequent analysis would reveal whether or not is had any statistical relevance.
- L. 179: How did you quantify the water balance error? Please explain.
- L. 190: The definition of outliers and their handling is not sound. Cook's distance is simply a mean of identifying data points that deviate in a statistical sense from the rest of the data population and have a strong influence on a derived regression model. This does not imply that the data point corresponds to an outlier that can be discarded from the analysis! It may be that the outlier reflects reality as well as all other data. They may simply reflect rare events. Of course it is important to assess the influence of statistical outliers on model performance and predictions. However, unless there are sound reasons to exclude data as outliers because these reasons indicate the outliers to be wrong, outliers have to be included in the analysis. For example, it can be made transparent that some data (explicitly shown) deviate from the others in a specific way and discuss possible reasons. Hiding them to the scientific community introduces bias.

**References**

Gassmann, Matthias, Jens Lange, and Tobias Schuetz. 2012. 'Erosion modelling designed for water quality simulation', Ecohydrology, 5: 269-78.
- Lefrancq, M., G. Imfeld, S. Payraudeau, and M. Millet. 2013. 'Kresoxim methyl deposition, drift and runoff in a vineyard catchment', Science of the Total Environment, 442: 503-08
- Schuetz, T., M. Weiler, and J. Lange. 2012. 'Multitracer assessment of wetland succession: Effects on conservative and nonconservative transport processes', Water Resources Research, 48: W06538

---

## Referee Comment (RC2) · Anonymous Referee #2 · 6 Aug 2020

The paper presents an interesting case study of the effect of vegetated treatment system to reduce contaminant inputs to surface waters. The topic is important, the field and lab work involved a lot of effort and the analysis can help to improve the understanding of contaminant loss and mitigation with vegetated treatment system. Therefore, it is worth publishing after additional clarification and correction. Overall, I have similar issues with the data analysis in the current state as reviewer 1. I support his remarks and think that they are very well stated. The main points to address as described below.

- The uncertainty in the target variables peak-concentration reduction rate and

mass removal rate are very high and largely ignored in the analysis. Some reasons for the uncertainties are:

- The contaminant concentrations are highly variable and the dynamics are difficult to capture with the applied measurement resolution. Important peaks can be missed and last data points do not always reflect baseflow concentrations. This missing information can lead to large errors in the peak-concentration reduction rates. Simple linear interpolation of concentration for mass calculation (L 157) can led to even higher bias for the mass removal rates. A flow proportional measurement could have been a better option. Was flow proportion measurements available?

- The timing of the last application of the investigated substances is completely neglected. However, more pronounced peaks are to be expected shortly after the application. Standardizing the concentrations or considering relative reduction rate does not completely solve this issue.

• I think a much better understanding of the uncertainty of the contamination measurement could be gained from a detailed analysis of the discharge behavior during the investigated events. This data is available in a much higher resolution (L86: Stream flow was measured every minute ...). Unfortunately, they are neither shown in detail nor really used in the analysis. It would be interesting to see the sample points/concentrations during an event together with the discharge measurement on a higher resolution, since the dynamics of the contaminants are driven mainly by the hydrology. I think this additional information would give an inside about how well the concentration dynamics have been captured. Moreover, information about the application patterns would improve the interpretation as well.

• The regression analysis is done rather poorly and the procedure is neither well explained nor well presented.

[Figure]

- It is not shown that the condition for a simple multiple linear regression are fulfilled, since the results are not validated or at least this information is not shown. I would at least expect a classical residual analysis in the supporting information.
- Automatically remove outliers based on a doubtful model without further analysis is not a proper way to go. For example, if outliers are a real problem, robust regression could be a solution (e.g. library robustbase).
- I don't think the requirement of independence of the data is fulfilled in this context. Data points of the same discharge event for the different components are not expected to be independent. Maybe a mixed model (with the discharge event as random effect) could help (e.g. library lme4). However, I doubt that more than a nice qualitative analysis of contaminate dynamics in a catchment with a vegetated treatment system will be possible with this setup.

- It is somehow obvious that dispersion has a stronger effect on substances with a more pronounced peak (as explained by reviewer 1).

- Although the clustering is done correctly, the connection with the discharge events is not well elaborated. Moreover, they are other clustering algorithms, which might be more robust (e.g. k-medoids. hierarchical Clustering). In L 207 it is written:"With the exception of cluster B which rather represented similar events (event 1 and event 4 in Fig. 2), overall clustering was controlled by similar behavior of contaminant groups." What was special by the event 1 and 4? Are these really exceptions? The contaminant groups seem to be important, however, I think the discharge dynamic and the application timing are important as well. Maybe it would also been interesting to cluster the discharge events. This data are also available in a higher resolution.

Detailed comments:

- L100: Fig. 2: I guess the discharge shown in Fig. 2 is from G1. This should be included in the description.

- L 108: Overall, herbicides have been also shown to be very persistence. For examples, atrazine has been detected after 10 years without application. (e.g. https://doi.org/10.1021/acs.est.7b02529)

- L 110: Azole pesticides are also persistent as indicated by many studies (e.g. https://doi.org/10.1016/j.envint.2020.105708)

- L 131: What is the accuracy and precision of the method? Has the analytical method validated?

- L145: Why is the cluster analysis important for the calculation of the dispersion sensitivity index? The index could also be calculated without clustering.

- L 191: From which mean? Do you mean 2 standard deviation from the prediction?

- L 205: It doesn't make sense to talk about a peak in Cluster C ("$T_{peak} = 6h$"). Not even the mean has a peak there.

- L 212: The surface runoff from the elevated vineyard has also to flow through the lower terrain slope to reach the river, expect that there are other shortcuts (streets, drains). See also reviewer 1).

- L 315: I do not understand the explanatory power for the different variable. Are they calculated by a univariate analyses? At least for me, the R-Output would be much easier to interpret.

---

## Author Comment (AC1) · 10 Sep 2020

**Response to comments by Referee #1**

We thank Anonymous Referee #1 for providing useful and constructive comments. We will carefully revise the manuscript and address all the points raised by the referee, as this will clearly improve the quality of our manuscript. Particularly, we intend to replace the linear regression model that was criticized by Referee #2 by a qualitative analysis.

The authors present an interesting study design with almost experimental character. Based on Fig. 1, one can conclude that there is a spatial separation between different crops in the catchment (arable crops versus vineyards). Because these crops receive different pesticide treatments (with regard to timing, compounds) one gets a signal in the outlet of the catchment that is related to a certain spatial unit. This design has the potential for learning how spatial aspects, physico-chemical properties of pesticides and agricultural practices influence pesticide losses (dynamics, loss rates). This could also be relevant for better understand the functioning of the wetland as is the main purpose of this manuscript.

Reply: We thank referee #1 for acknowledging the unique character of our experimental setup to investigate different types of contaminant input signals and their mitigation in an agricultural wetland. We will highlight these aspects more clearly in the updated version of our manuscript.

The manuscript analyses the effect of a wetland at the outlet of this agricultural catchment on pesticide transfer further downstream. Based on samples taken during 10 events and five baseflow periods, the retention capacity is studied for four active ingredients and two metabolites regarding the relative reduction of peak concentrations and pesticide mass transported downstream. Specifically, the authors investigated the effect of the shape of the chemograph at the inlet and a few compound properties and explain the differences in the chemograph shapes of different compounds by being applied to different parts of the catchment.

Monitoring pesticide dynamics in natural systems is demanding and requires substantial efforts in the lab and in the field. The observed pesticide behaviour in the environment is complex and often influenced by specific conditions at the local scale. Therefore it is valuable for science and practice if such observations get published from different locations and catchments because it broadens our understanding of the environmental fate of pesticides under different conditions.

Unfortunately, there a number of issues that limit the value of the manuscript in its current status. I describe my main concerns below and add some further details at the end.

**Scientific quality:**
The data analysis is rather superficial and several of important conclusions are not strongly backed up with data. This holds true for example for the interpretation of the different dynamic patterns identified (L. 210 - 221). For example, the authors hypothesise that fungicides from the upstream vineyards are more quickly transported to the stream than the herbicides applied closer to the creek in the valley bottom (L. 210 - 213). Why this should be the case remains unclear. Checking a previous publication on the catchments (Gassmann et al., 2012) reveals that there is a dense road network connected to the pipe drains. These structures are made explicit in the earlier publication but no linear structures are indicated in the catchment map of this manuscript. Given the fact that pesticide drift to non-target surfaces such as roads may be important for pesticide transport in vineyards (Lefrancq et al., 2013) important aspects of the catchment are neglected and not included into the discussion of the results.

Reply: We agree that structures like roads may represent shortcuts and accelerate pesticide transport. Particularly spray drift is an important process here, which is more relevant for fungicides, as they are applied into the foliage and not close to the ground as herbicides. We are thankful for this hint and will discuss this in detail in the revised manuscript.

Also temporal aspects are only treated superficially. The shape of the chemograph of a pesticide is strongly influenced by the rainfall dynamics. Because the authors compare the dynamic of different pesticides across different events (this is not really evident from the main text, but see the SI), differences in concentration dynamics could also be strongly influenced by rainfall patterns and discharge behaviour. Unfortunately, no respective data is shown or discussed.

Reply: Description of the experimental setup and the included data will be revised to make it clearer. A figure containing concentrations and discharge during all events will be provided and effects on chemographs will be discussed.

This holds also true for the timing between the last application of a pesticide and the rainfall event. The authors do not discuss this aspect and treat all compounds and all events the same (except for two events for flufenacet where too many samples $<$ LOQ). However, inspection of the actual concentration data reveal strong differences in the concentration levels of the different compounds across all events. High concentrations of several hundred to thousands of ng $L^{-1}$ were found for boscalid for all events, while flufenacet was only found in one event above 1600 ng $L^{-1}$ but otherwise never above 40 ng $L^{-1}$. Obviously, the history of the compounds since the last application was very different. Neglecting such aspects but interpreting the different relative concentration dynamics with respect to transport differences from different parts of the catchments is not very solid.

This also limits the value of the cluster analysis. The results in Fig. 3 show that a given compound appears in different clusters. However, this aspect is not properly discussed and no explanation is provided why this was the case nor what this implies for interpretation of these clusters.

Reply: We fully agree that the chemograph shape is influenced by many factors including compound properties, source areas, transport pathways, absolute concentrations, rainfall and discharge dynamics and also pesticide application time, some of which we have not addressed

to the necessary level of detail so far. In general, we see three groups of factors that may influence chemographs. These are compound properties (e.g. sorption affinity, degradability), event properties (rainfall and discharge dynamics) and catchment properties (application areas, rates and timing, transport pathways, slopes). In the investigated catchment, different pesticide types are applied to different compartments of the catchment so that differentiation between compound and catchment-related factors is challenging. However, we consider our analysis valuable as it revealed that (1) systematic differences were evident in chemographs and (2) clustering was mainly according to compound types, rather than according to events. This suggests that the compound type (and co-occurring catchment-related factors, e.g. source areas and transport pathways, application rate and time) had larger influence on chemograph shapes than event properties. We are thankful to the reviewer, since obviously we did not make these points clear enough and will discuss them in more detail in a revised manuscript.

Additionally, it is not clear to which degree these results reflect the full spectrum of observed pesticide dynamics. The authors mention in the Method section (L. 190) that they have removed outliers based on purely statistical analyses. I don't think the procedure is sound (see below) and may bias the findings by excluding unusual - or simply rare - dynamics.

Reply: No outliers were removed for the cluster analysis so that this issue is limited to the regression model. As we intend to replace the linear regression model by a qualitative data analysis in response to objections by referee #2, we consider this point obsolete.

Finally, the calculation retention rates raises a number of question marks, which may further impact the subsequent linear model for describing the retention efficiency of the wetland. First, given the measured concentrations (see SI, but also Fig. 3) it is evident that in many cases the last data point does not reflect baseflow concentrations after the event. Accordingly, the mass loss during the events may have been substantially larger in some cases. The extent of this effect depends on the unobserved concentration dynamics but also on the discharge. Unfortunately, no discharge data is provided that illustrate which part of the event hydrographs have actually be covered by the sampling.

Reply: We are aware of the issue with elevated concentrations in the last event sample and calculated mass flux during the events in order to assess the effect on mass balances ($R_M$). Despite elevated concentrations, mass flux was close to zero at the end of sampling in most cases. We agree that this information should be accessible to the reader and will provide a figure showing mass fluxes in the supplementary material of a revised manuscript. Late time concentration was largely irrelevant for $R_C$ as peaks normally occurred in earlier samples. $R_C$ may, however, be influenced by discharge dynamics. A figure for this will be provided in the revised manuscript so that the reader obtains an idea on associated uncertainties.

Second, the observed concentration levels demonstrate that for compounds such as flufenacet most events reflect conditions long after the last application. Accordingly, the concentration signal is rather weak and calculated retention is laced with considerable uncertainty. This aspect is not mentioned.

Reply: We agree that relative uncertainty is higher at lower concentration and will include this aspect into the discussion of a revised manuscript.

**Scientific significance**:

The issues listed above reduce the scientific significance of the manuscript. Additionally, there are questions about the scientific insight conveyed by the manuscript. There are two major issues:

**Relevance of the shape of the chemograph** The effect of the chemograph is almost a trivial finding. Given the short residence time in the wetland (about 1 h; see L. 81) degradation processes will be very limited and the main effect on peak concentrations in the outlet is expected by dispersion (as expected by the authors, see below), i.e. mixing water of different concentrations at the inlet. Obviously, the more variable the input, the larger the relative effect of mixing on the relative maximum concentrations. Actually, the authors have stated this outcome already in the Introduction as fact: "'*Peak concentration reduction will be stronger for a signal with pronounced peak and low background than for a signal with small peak and high background if both signals are exposed to the same dispersion.*'" (L. 54 - 55).

> [... For mathematical derivation by referee #1 see original comment …]

Reply: We agree that the finding that different signals are affected differently by dispersion is not novel and we did not claim that it is, although we are not aware of many wetland studies that explicitly address this aspect. Instead, contaminant peak reduction is often shown as an important mitigation effect of wetlands, as it decreases acute toxicity (Bundschuh et al., 2016; Elsaesser et al., 2011). However, this is only one aspect of the problem, because peak reduction does not necessarily mean that contaminant mass is also reduced. In our study (sampling scheme, data analysis, etc.) we are addressing both types of toxicity (acute and permanent). We are aware that this aspect was not communicated clearly enough in our manuscript and we will revise our updated version correspondingly.

> In this context I am furthermore surprised that the authors did not use the model that was developed for the study wetland for evaluating the effect of different chemograph shapes (Schuetz et al., 2012). Although the wetland may have undergone changes since the last tracer experiments, it would be a useful null model for testing how different input signals influence the reduction in peak concentrations.

Reply: We thank referee #1 for this suggestion and agree that a proper transport model would be useful. In fact, we experimented quite a lot with possible representations of the investigated system in the solute transport model (OTIS) used by Schuetz et al. (2012). Although we were actually able to reproduce concentrations at the basin outlet from those at the inlet acceptably well, our confidence in the model was low and we decided to not include the model for the following reasons:

(1) For evaluation of different input signals, it is crucial that solute transport (incl. dispersion) was well parameterized. OTIS was designed for stationary flow-conditions. Application of OTIS during transient flow is technically possible. However, we consider the parameterization in such cases questionable as model parameters that are obviously time-variant have to be assumed constant, e.g. storage zone area,

dispersion coefficient, and exchange rate. Although a conservative tracer injection was performed during one of the discharge events, we were unable to identify a range of transport parameters that was plausible when compared to the discharge conditions during the events.

(2) Parameterization of transport clearly influenced process-related parameters, i.e. rates of decay and kinetic sorption, so that the latter could hardly be estimated from the model.

(3) Comparison with Schuetz et al. (2012) was not possible because their model was based on stationary discharge, while we were dealing with event data and the studied system was fundamentally changed by the implementation of the retention pond in 2016, between the experiments by Schuetz et al. (2012) and the start of sampling for the present study.

Therefore we came to the conclusion that the use of an OTIS-type model and the interpretation thereof rather introduced additional uncertainties and that further insights provided by the model were limited.

**How to generalise the findings?** It is difficult to generalise any of the findings reported in the manuscript because they are very context dependent and results such as the cluster analysis are very phenomenological. The aspect that the shape of the chemograph has an influence on the relative degree by which peak concentrations are reduced is quite evident for the type of system under investigation.

Reply: We agree that findings from field experiments often depend on local conditions. However, regarding processes in the catchment, we do not consider the results of the cluster analysis "very phenomenological". We showed that the chemograph shape more strongly depended on catchment or compound properties and not on event characteristics. Potentially influential factors that can be separated by this type of analysis may be different in other catchments, e.g. source areas and compound properties might be distinguished more easily if application areas of the compound groups were not spatially separated as in this study. We therefore see a high potential for this type of analysis in other catchments, although local conditions have to be considered. We thank referee #1 for this objection as it shows that we did not communicate this aspect clearly enough. We will revise our manuscript accordingly.

We intentionally focus on the chemograph shape as it links processes in catchments to those in treatment wetlands and we consider this aspect in principle transferable to other systems. Although the a larger dispersion of sharp peaks is not novel, the importance of this relationship is generally not reflected in existing literature of contaminant mitigation so that we consider our work a valuable contribution to the body of knowledge in this field.

**Detailed comments:**

L. 30: The phrase "'... which may be equally or more mobile, persistent and toxic than their PC ...'" is misleading because it does not mention the general case that transformation products are less toxic.

Reply: This will be stated more clearly in a revised manuscript.

L. 93: How often were grab samples taken?

Reply: Grab samples were taken at 7 occasions, after careful inspection of the hydrograph, we conclude that 5 of the sampling in fact represented stationary flow conditions. This will be stated more clearly in a revised manuscript.

L. 123: How adequate is it to only take one single isotope-labelled internal standard not corresponding to the target compounds? Checking these compounds in one of our current analytical methods, retention times vary between 16.7 (metazachlor-ESA) and 21.0 min (penconazole). Also the KOC values vary by a factor of 400 between these two compounds. Please provide additional information supporting the assumption that terbutryn as an adequate internal standard (e.g., recoveries).

Reply: We will provide information accordingly in the manuscript.

L. 129 - 130: Please check the correct number of significant digits (can you measure with a precision of tens of picograms per litre?).

Reply: We will re-check the precision of our measurements and adopt the number of digits.

L. 142: Please provide the version of R. I assume that you did not implement the algorithm but used `kmeans()` implemented in standard R.

Reply: The version of R will be provided.

L. 174 - 176: Why did you include DT50 a priori? I'd recommend to leave it in. The subsequent analysis would reveal whether or not is had any statistical relevance.

Reply: As we intend to replace the linear regression model by a qualitative data analysis in response to objections by referee #2, we consider this point obsolete.

L. 179: How did you quantify the water balance error? Please explain.

Reply: What we called the water balance error is the relative difference between water masses registered at both gauges G1 and G2. We acknowledge that this term may be misleading as it does not necessarily represent a measurement error. We will call this parameter "Relative water balance anomaly" in a revised manuscript and describe how it was calculated.

L. 190: The definition of outliers and their handling is not sound. Cook's distance is simply a mean of identifying data points that deviate in a statistical sense from the rest of the data population and have a strong influence on a derived regression model. This does not imply that the data point corresponds to an outlier that can be discarded from the analysis! It may be that the outlier reflects reality as well as all other data. They may simply reflect rare events. Of course it is important to assess the influence of statistical outliers on model performance and predictions. However, unless there are sound reasons to exclude data as outliers because these reasons indicate the outliers to be wrong, outliers have to be included in the analysis. For example, it can be made transparent that some data (explicitly

shown) deviate from the others in a specific way and discuss possible reasons. Hiding them to the scientific community introduces bias.

Reply: We acknowledge that we should have been more transparent about handling of outliers in the manuscript and will carefully address this point in a revised manuscript. Outliers were only excluded in the regression model but not in the cluster analysis.

**References**

Bundschuh, M., Elsaesser, D., Stang, C., and Schulz, R.: Mitigation of fungicide pollution in detention ponds and vegetated ditches within a vine-growing area in Germany, Ecol. Eng., 89, 121–130, doi:10.1016/j.ecoleng.2015.12.015, 2016.

Elsaesser, D., Blankenberg, A.-G. B., Geist, A., Mæhlum, T., and Schulz, R.: Assessing the influence of vegetation on reduction of pesticide concentration in experimental surface flow constructed wetlands: Application of the toxic units approach, Ecol. Eng., 37, 955–962, doi:10.1016/j.ecoleng.2011.02.003, 2011.

---

## Author Comment (AC2) · 10 Sep 2020

**Response to comments by Referee #2**

We thank Anonymous Referee #2 for her/his for providing useful and constructive comments. We will carefully revise the manuscript and address all the points raised by the Referee. Particularly, we intend to replace the linear regression model that was criticized by referee #2 by a qualitative analysis.

The paper presents an interesting case study of the effect of vegetated treatment system to reduce contaminant inputs to surface waters. The topic is important, the field and lab work involved a lot of effort and the analysis can help to improve the understanding of contaminant loss and mitigation with vegetated treatment system. Therefore, it is worth publishing after additional clarification and correction. Overall, I have similar issues with the data analysis in the current state as reviewer 1. I support his remarks and think that they are very well stated. The main points to address as described below.

- The uncertainty in the target variables peak-concentration reduction rate and mass removal rate are very high and largely ignored in the analysis. Some reasons for the uncertainties are:

  - The contaminant concentrations are highly variable and the dynamics are difficult to capture with the applied measurement resolution. Important peaks can be missed and last data points do not always reflect baseflow concentrations. This missing information can lead to large errors in the peak-concentration reduction rates. Simple linear interpolation of concentration for mass calculation (L 157) can led to even higher bias for the mass removal rates. A flow proportional measurement could have been a better option. Was flow proportion measurements available?

Reply: We agree with referee #2 that flow proportional sampling is the preferred option if accurate mass balances are desired. As already stated in our response to the first referee we aimed at including both aspects (peak and mass reduction) of pesticide mitigation in our approach. As such we chose flow-triggered sampling at fixed times with dense sampling at the beginning and increasing intervals towards later times. As in any sampling strategy (limitation of resources) there is a tradeoff between number of samples per event and number of events to be sampled. We evaluate our approach as a meaningful compromise. However, we further agree that uncertainty is of this procedure is difficult to evaluate. We will include this problem in the discussion section.

  - The timing of the last application of the investigated substances is completely neglected. However, more pronounced peaks are to be expected shortly after the application. Standardizing the concentrations or considering relative reduction rate does not completely solve this issue.

Reply: We agree that detailed information on application rates would be beneficial for interpretation of the pesticide signals emerging from the catchment. We will discuss this aspect in more detail in a revised manuscript (also see reply to referee #1). We generally think that the timing of application is one of several factors that affect relative reduction rates via chemograph shape which is the key point of our study.

- I think a much better understanding of the uncertainty of the contamination measurement could be gained from a detailed analysis of the discharge behavior during the investigated events. This data is available in a much higher resolution (L86: Stream flow was measured every minute ...). Unfortunately, they are neither shown in detail nor really used in the analysis. It would be interesting to see the sample points/concentrations during an event together with the discharge measurement on a higher resolution, since the dynamics of the contaminants are driven mainly by the hydrology. I think this additional information would give an inside about how well the concentration dynamics have been captured. Moreover, information about the application patterns would improve the interpretation as well.

Reply: We will provide a figure showing all events and compounds together with discharge dynamics in the revised manuscript and discuss this aspect.

- The regression analysis is done rather poorly and the procedure is neither well explained nor well presented.

  - It is not shown that the condition for a simple multiple linear regression are fulfilled, since the results are not validated or at least this information is not shown. I would at least expect a classical residual analysis in the supporting information.

Reply: We thank the reviewer for this point. We actually did perform a residual analysis in which we did not find indication for violation of the assumptions of multiple linear regression. The results should have been communicated in the original manuscript. However, as we intend to drop the regression model as a consequence to further objections (s. below) we consider this point obsolete.

  - Automatically remove outliers based on a doubtful model without further analysis is not a proper way to go. For example, if outliers are a real problem, robust regression could be a solution (e.g. library robustbase).

Reply: In the original manuscript we removed two outliers in the RC and one in the RM model based on Cook's distance and standard deviation with the intention to reduce the influence of high leverage points. We agree that we should have dealt more sensitively with this issue and will revise the handling of outliers, i.e. not remove data points in a qualitative analysis.

– I don't think the requirement of independence of the data is fulfilled in this context. Data points of the same discharge event for the different components are not expected to be independent. Maybe a mixed model (with the discharge event as random effect) could help (e.g. library lme4). However, I doubt that more than a nice qualitative analysis of contaminate dynamics in a catchment with a vegetated treatment system will be possible with this setup.

Reply: We thank referee #2 for this interesting objection to the model used in our study and realize that we should have more carefully considered the structure of our data. Although other studies likewise neglected this aspect (Bundschuh et al., 2016; Stehle et al., 2011), we agree that a mixed model would be a better option. The intention of the regression analysis was to facilitate a comparison of model parameters identified as influential with the results of other studies. Now that we realize that we realize that the results of such other studies should also be treated with caution, the statistical analysis becomes somewhat obsolete. We therefore decide to go with the recommendation by referee #2 and limit ourselves to a qualitative analysis in which we compare potentially influential variables to reduction rates of peak concentration and compound mass. Boxplots will be used to illustrate variability in events for data available on the level of compounds and vice versa.

• It is somehow obvious that dispersion has a stronger effect on substances with a more pronounced peak (as explained by reviewer 1).

Reply: We agree with referee #2 that it is obvious that dispersion affects well-defined peaks stronger than flattened signals. However, as already discussed in our response to reviewer #1, it is not adequately stressed in existing literature on wetland contaminant mitigation.

• Although the clustering is done correctly, the connection with the discharge events is not well elaborated. Moreover, they are other clustering algorithms, which might be more robust (e.g. k-medoids. hierarchical Clustering). In L 207 it is written:"With the exception of cluster B which rather represented similar events (event 1 and event 4 in Fig. 2), overall clustering was controlled by similar behavior of contaminant groups." What was special by the event 1 and 4? Are these really exceptions? The contaminant groups seem to be important, however, I think the discharge dynamic and the application timing are important as well. Maybe it would also been interesting to cluster the discharge events. This data are also available in a higher resolution.

Reply: We agree with referee #2 that similar chemographs may emerge for many reasons, including compound-related (sorption affinity, degradability, application rate and timing), catchment-related (transport pathways, application areas) or event-specific (amount and dynamics of rainfall and subsequent discharge, incl. multiple peaks and spatial heterogeneity in rainfall) factors. As clustering is solely based on measured concentrations, the resulting clusters

are independent of whether we are aware of all relevant processes or not. We found that cluster A, C, and D mainly differed according to compound-related or catchment-related factors, separation of which is a bit challenging due to spatially separated application areas of fungicides and herbicides in the studied catchment (see also our comments to reviewer #1 above). In contrast, cluster B mainly reflected two different events. Examination of the discharge dynamics during the two events in cluster B did not reveal any obvious peculiarities. What made the two events in cluster B special was the absence of a time lag between peaks of fungicides and herbicides (and their TPs) that was usually observed and reflected in clusters A and D. The quicker response of herbicides in cluster B may be due to a recent application, however, exact application times and rates are unknown. We will include these observation in our updated manuscript.

We thank referee #2 for the suggesting to also consider alternative clustering algorithms. We agree that cluster centroids in k-medoids are more robust against outliers than cluster centers in k-means and revised our analysis correspondingly. We found similar clusters as when using k-means. In fact, partitioning between fungicides and other compounds was slightly better.

We thank referee # 2 for the suggestion to also cluster discharge events and agree that this may be helpful to check whether similar discharge events produced similar chemographs. However, we think that the number of discharge events with pesticide data is too low (n=10) for such an analysis. We will therefore rather include discharge conditions into comparison of cluster properties results of which will be included into the qualitative analysis of pesticide mitigation in the wetland as described above. We will make the rationale of the cluster analysis clearer and discuss the interpretation of the clusters in more detail in a revised manuscript.

**Detailed comments**

L100: Fig. 2: I guess the discharge shown in Fig. 2 is from G1. This should be included in the description.

Reply: This information will be added to the figure capture.

L 108: Overall, herbicides have been also shown to be very persistence. For examples, atrazine has been detected after 10 years without application. (e.g. https://doi.org/10.1021/acs.est.7b02529)

Reply: This was not meant to be a general statement. These lines (also s. next comment) just sum up, how the selected compounds are classified according to the Pesticide Properties Data Base (Lewis et al., 2016). We will make this more clear in a revised manuscript.

L 110: Azole pesticides are also persistent as indicated by many studies (e.g. https://doi.org/10.1016/j.envint.2020.105708)

Reply: (s. above)

L 131: What is the accuracy and precision of the method? Has the analytical method validated?

Reply: We will provide additional information about the method in a revised manuscript.

L 145: Why is the cluster analysis important for the calculation of the dispersion sensitivity index? The index could also be calculated without clustering.

Reply: We acknowledge that this phrasing was imprecise. The clustering revealed that there were major differences in chemograph shapes, particularly regarding the tailings of the breakthroughs. Thus, the clusters provided the idea for the index which is then considered a way to integrate the latter observation into further analysis.

L 191: From which mean? Do you mean 2 standard deviation from the prediction?

Reply: We acknowledge that our procedure for identification was debatable. As mentioned above we will change our analysis so that outliers do not have to be removed.

L 205: It doesn't make sense to talk about a peak in Cluster C ("T peak = 6h"). Not even the mean has a peak there.

Reply: This is true and will be corrected in a revised manuscript.

L 212: The surface runoff from the elevated vineyard has also to flow through the lower terrain slope to reach the river, expect that there are other shortcuts (streets, drains). See also reviewer 1).

Reply: We thank referee #2 for this comment as it points out that we have to consider the role of the catchment in more detail. We will revise our manuscript accordingly.

L 315: I do not understand the explanatory power for the different variable. Are they calculated by a univariate analyses? At least for me, the R-Output would be much easier to interpret.

Reply: The explanatory power of the resulted from decomposition of total R2 according to the method of Grömping (2006) (L. 193) which uses the mean over all possible orders of parameter addition to multivariate models. However, this will no longer be relevant as the regression model is discarded.

**References**

Bundschuh, M., Elsaesser, D., Stang, C., and Schulz, R.: Mitigation of fungicide pollution in detention ponds and vegetated ditches within a vine-growing area in Germany, Ecol. Eng., 89, 121–130, doi:10.1016/j.ecoleng.2015.12.015, 2016.

Grömping, U.: Relative importance for linear regression in R: The package relaimpo, Journal of Statistical Software, 17, 2006.

Lewis, K. A., Tzilivakis, J., Warner, D. J., and Green, A.: An international database for pesticide risk assessments and management, Hum. Ecol. Risk Assess., 22, 1050–1064, doi:10.1080/10807039.2015.1133242, 2016.

Stehle, S., Elsaesser, D., Gregoire, C., Imfeld, G., Niehaus, E., Passeport, E., Payraudeau, S., Schäfer, R. B., Tournebize, J., and Schulz, R.: Pesticide risk mitigation by vegetated treatment systems: a meta-analysis, Journal of environmental quality, 40, 1068–1080, doi:10.2134/jeq2010.0510, 2011.

---

## Author Response (AR1)

Dear Dr. Zehe,

We are thankful for the constructive comments by the editor and both referees which helped us to considerable improve the quality of our manuscript. Below please find our revised manuscript with pointby-point answers to the comments of the referees. In the document below referee comments are represented in black and our responses in blue. Italic font indicates quotations from the revised manuscript. Line numbers refer to the mark-up version of the manuscript below.

**Response to comments by Referee #1**

The authors present an interesting study design with almost experimental character. Based on Fig. 1, one can conclude that there is a spatial separation between different crops in the catchment (arable crops versus vineyards). Because these crops receive different pesticide treatments (with regard to timing, compounds) one gets a signal in the outlet of the catchment that is related to a certain spatial unit. This design has the potential for learning how spatial aspects, physico-chemical properties of pesticides and agricultural practices influence pesticide losses (dynamics, loss rates). This could also be relevant for better understand the functioning of the wetland as is the main purpose of this manuscript.

The manuscript analyses the effect of a wetland at the outlet of this agricultural catchment on pesticide transfer further downstream. Based on samples taken during 10 events and five baseflow periods, the retention capacity is studied for four active ingredients and two metabolites regarding the relative reduction of peak concentrations and pesticide mass transported downstream. Specifically, the authors investigated the effect of the shape of the chemograph at the inlet and a few compound properties and explain the differences in the chemograph shapes of different compounds by being applied to different parts of the catchment.

Monitoring pesticide dynamics in natural systems is demanding and requires substantial efforts in the lab and in the field. The observed pesticide behaviour in the environment is complex and often influenced by specific conditions at the local scale. Therefore it is valuable for science and practice if such observations get published from different locations and catchments because it broadens our understanding of the environmental fate of pesticides under different conditions.

Unfortunately, there a number of issues that limit the value of the manuscript in its current status. I describe my main concerns below and add some further details at the end.

**Scientific quality:**

The data analysis is rather superficial and several of important conclusions are not strongly backed up with data. This holds true for example for the interpretation of the different dynamic patterns identified (L. 210 - 221). For example, the authors hypothesise that fungicides from the upstream vineyards are more quickly transported to the stream than the herbicides applied closer to the creek in the valley bottom (L. 210 - 213). Why this should be the case remains unclear. Checking a previous publication on the catchments (Gassmann et al., 2012) reveals that there is a dense road network connected to the pipe drains. These structures are made explicit in the earlier publication but no linear structures are indicated in the catchment map of this manuscript. Given the fact that pesticide drift to non-target surfaces such as roads may be important for pesticide transport in vineyards (Lefrancq et al., 2013) important aspects of the catchment are neglected and not included into the discussion of the results.

Reply: We agree that structures like roads may represent shortcuts and accelerate pesticide transport. We improved the description of the catchment accordingly, added the primary drainage network to the map of the catchment (Figure 1), and discussed different mobilization dynamics in more detail. The description of the now reads as follows:

"The study site (Figure 1) is located inside a flood detention basin in the 1.8 km2 Loechernbach catchment, southwest Germany. Catchment elevation ranges between 213 m.a.s.l. at the outlet and 378 m.a.s.l. in the western corner. Mean precipitation was 800 mm a-1 between 2009 and 2018, mean air temperature 11.3 °C. Soils mainly consist of calcaric regosols which formed on aeolian loess and have a typical grain size distribution of 10 % sand, 80 % silt and 10 % clay. Most of the catchment is dedicated to large artificial vineyard terraces (71%), while croplands occupy the valley bottoms (20%). Forest only accounts for a small portion (9%) and is limited to the most elevated part of the catchment. This partition in land use is reflected in the main application areas of pesticide types. Fungicides are applied on vineyard terraces, while herbicides are mainly applied to the cropland in the flat valleys. Large parts of the catchment are drained by a sub-surface pipe network (Figure 1) connecting vineyards and paved roads to the main channel in the valley. This drainage network causes fast downstream transport of storm water and suspended sediments (Gassmann et al., 2012). In addition, fields in the valley bottoms are drained by a secondary network of smaller and usually shorter field drains that are either connected to the primary drainage network or directly connected to the stream (Schuetz et al., 2016). A 20,000  $m^3$  detention basin was built at the outlet of the Loechernbach to prevent flooding of the downstream village." (Lines 98-112)

More emphasis is now laid on hydrological short cuts in the interpretation of the chemograph clusters, particularly regarding cluster B:

"The fact that concentration in cluster B quickly increased with discharge (within 30 minutes) is in line with fast transport from the vineyard terraces to the stream via roads and drainage pipes as described by Gassmann et al. (2012) for suspended solids in the studied catchment. Along such preferential pathways, compound properties, such as sorption affinity, may be less important (Gomides Freitas et al., 2008) compared to e.g. percolation through the soil with intense contact to sorption sites in the soil matrix. Moreover, fungicides are applied by sprayers into the foliage and can drift to e.g. paved surfaces from which they can be quickly mobilized by subsequent rainfall (Lefrancq et al., 2013). We therefore hypothesize that cluster B was mainly produced by surface flushing and fast transport pathways of fungicides. This explained the quick rise and subsequent decline in concentrations (concurrent with plateaus produced by slower flow components in cluster A)." (Lines 574-583)

Also temporal aspects are only treated superficially. The shape of the chemograph of a pesticide is strongly influenced by the rainfall dynamics. Because the authors compare the dynamic of different pesticides across different events (this is not really evident from the main text, but see the SI), differences in concentration dynamics could also be strongly influenced by rainfall patterns and discharge behaviour. Unfortunately, no respective data is shown or discussed.

Reply: We improved the description of the overall aim of this study:

"We hypothesize that the efficiency of contaminant mitigation in VTSs depends on the shape of the input chemographs and eventually on the factors that produce these signals in the catchment. In order to test this hypothesis we grouped chemographs of 6 contaminants mobilized in a viticultural catchment during 10 discharge events into clusters according to chemograph shape. We then compared peak concentration reduction and mass removal in a downstream VTS both among clusters and in terms of compound properties and discharge dynamics." (Lines 74-78)

Different discharge dynamics of the events are now described qualitatively:

"The 10 events were characterized by different discharge magnitudes and dynamics (Figure 3). Mean discharge during the events ranged between 0.7 (E10) and 32.0 L s-1 (E2) with respective peak values between 4.4 (E10) and 199.7 L s-1 (E2). The recorded event hydrographs included events with one single discharge peak (E4, E5, E6, E10), with one major peak followed by one or more secondary peaks (E2, E3, E7, E9), and events in which a major peak followed an earlier smaller peak (E1, E8). In most cases discharge had recessed to pre-event levels by the end of the 12-hour sampling procedure, only E1 and E2 showed ongoing flow recession." (Lines 330-337) The influence of discharge dynamics on chemograph shapes was added to the discussion of chemograph clusters, particularly of cluster D:

"Cluster D included chemographs of both herbicides and TPs and presented a clear peak that was often defined by a single sample 2 h after the beginning of the event. In contrast to cluster A, cluster D was characterized by a single sharp discharge peak (except in event E7 where a second peak occurred shortly after the first) and mainly included chemographs during periods of low flow. Our interpretation is that cluster D represented flow events in which no dilution of herbicide and TP fluxes by fungicide fluxes or secondary discharge peaks occurred. Low pre-event discharge in cluster D compared to cluster A may indicate low water levels which may have caused a slower response as no enriched pre-event water was released from the soils in the valleys." (Lines 594-600)

In order to provide a better idea of our data set, we replaced Figure 2 in the original manuscript, by a new figure (Figure 3) showing discharge dynamics and pesticide concentrations during all 10 events both at the inlet and the outlet of the detention system.

This holds also true for the timing between the last application of a pesticide and the rainfall event. The authors do not discuss this aspect and treat all compounds and all events the same (except for two events for flufenacet where too many samples < LOQ). However, inspection of the actual concentration data reveal strong differences in the concentration levels of the different compounds across all events. High concentrations of several hundred to thousands of ng L-1 were found for boscalid for all events, while flufenacet was only found in one event above 1600 ng L-1 but otherwise never above 40 ng L-1. Obviously, the history of the compounds since the last application was very different. Neglecting such aspects but interpreting the different relative concentration dynamics with respect to transport differences from different parts of the catchments is not very solid.

This also limits the value of the cluster analysis. The results in Fig. 3 show that a given compound appears in different clusters. However, this aspect is not properly discussed and no explanation is provided why this was the case nor what this implies for interpretation of these clusters.

Reply: We fully agree that the chemograph shape can be influenced by many factors and our analysis is not able to account for all of them. We do not have data on exact pesticide application times and rates, but included this aspect in the discussion. Given the spatial separation of fungicide and herbicide application areas, it seems highly plausible that these groups of compounds show different transport dynamics. In the revised manuscript we discussed the chemograph clusters and factors that cause their differentiation in more detail. The potential effect of recent application was particularly addressed with respect to cluster C:

"Cluster C was composed of chemographs of all compounds but mainly from events E2 and E4 indicating event dependence. Two aspects were found to support this idea. First, there was a secondary discharge peak in event 2 that did not contribute much in terms of contaminant concentration but rather caused dilution and produced particularly flat chemograph tails. Second, peaks of herbicides and TPs were less delayed compared to fungicides. This may be the result of recent herbicide application and active surface runoff in the flat valleys. Timing of pesticide application was identified as the main export driver of currently used pesticides by Imfeld et al. (2020) who performed a cluster analysis on rainfall data from a headwater vineyard catchment. Based on the magnitude of discharge and amount of mobilized contaminants (concentration of metazachlor  $\approx 10 \ \mu g \ L^{-1}$ ), both explanations seem plausible in event E2. Event E4, however, did not show particularly high herbicide concentration nor a secondary discharge peak. Although it is obvious that chemograph shapes in cluster C differed from the other clusters, unfortunately, the responsible factors remain unclear." (Lines 584-593)

We also added a paragraph addressing the effect of overlooking important aspects on the interpretation of the clusters:

"The unclear interpretation of cluster C suggest that we missed important factors for the formation of chemographs. In fact, variables like spatial distribution of rainfall or pesticide application rates and timing (Imfeld et al., 2020) and possibly other factors likely influenced chemograph shapes. Knowing all these variables would not change the results produced by the clustering algorithm but rather increase our ability to interpret them. Nevertheless, the cluster analysis helped to explore how the catchment and processes therein influenced concentration signals of mobilized contaminants." (Lines 601-608)

Additionally, it is not clear to which degree these results reflect the full spectrum of observed pesticide dynamics. The authors mention in the Method section (L. 190) that they have removed outliers based on purely statistical analyses. I don't think the procedure is sound (see below) and may bias the findings by excluding unusual - or simply rare - dynamics.

Reply: Following referee #2 (see below) we excluded the regression model for which we had removed outliers. In the present analysis no outliers were removed.

Finally, the calculation retention rates raises a number of question marks, which may further impact the subsequent linear model for describing the retention efficiency of the wetland. First, given the measured concentrations (see SI, but also Fig. 3) it is evident that in many cases the last data point does not reflect baseflow concentrations after the event. Accordingly, the mass loss during the events may have been substantially larger in some cases. The extent of this effect depends on the unobserved concentration dynamics but also on the discharge. Unfortunately, no discharge data is provided that illustrate which part of the event hydrographs have actually be covered by the sampling. Reply: see below.

Second, the observed concentration levels demonstrate that for compounds such as flufenacet most events reflect conditions long after the last application. Accordingly, the concentration signal is rather weak and calculated retention is laced with considerable uncertainty. This aspect is not mentioned.

Reply: We added a section to the discussion dealing with uncertainties of the monitoring setup and added a figure to the supplementary material showing that contaminant mass flux was usually very low at the end of the sampling procedure (Figure S1):

"Regarding chemographs and calculation of  $R_{\rm C}$ , uncertainties arose from timing and frequency of sampling and analytical error, and additionally from discharge measurement when calculating masses and  $R_M$ . Analytical methods used in this study usually produced very consistent results so that variability in concentrations of parent compounds in duplicate samples was low (sd < 10 %). However, in individual samples collected at G1 analytical variability was elevated for met-ESA and met-OA (Figure 3), reducing confidence in concentrations and the derived measures  $R_C$  and  $R_M$  of TPs in the affected chemographs (E2, E5, E7, E8, E9). Uncertainty related to timing and frequency of sampling can hardly be quantified but certainly depends on how well the sampling intervals captured variability in concentrations during flood events and how well the time lag between upstream and downstream sampling matched the residence time of solutes in the wetland. Lefrancq et al. (2017) assessed the effect of sampling frequency in pesticide monitoring data collected during runoff from a single vineyard and found that acute toxicity of pesticide flushes was underestimated up to 4-times when calculated from event means and up to 30-times when calculated from random samples. Although these data were collected on the plot scale and we assume that variability in our catchment is lower due to longer flow paths and mixing processes on the catchment scale, uncertainty of the chemographs in our study could have been reduced by increasing sampling frequency. Regarding the timing of upstream and downstream sampling, there is evidence that water residence time in the wetland was in fact shorter than one hour. The observation that for quickly responding compounds, such as boscalid, concentration in the first sample at G2 was often elevated compared to the first sample at G1 indicates that the contaminant flush hat already reached G2 when sampling started. This did not influence determination of  $C_{out,max}$  and  $R_C$  in the outlet of the wetland, as concentrations were still rising from the first to the second sample (Figure 3). However, effects on  $M_{out}$  were higher, since a relevant fraction of contaminant mass leaving the wetland was not registered and thereby caused overestimation of  $R_M$  (Figure S1). Another source of uncertainty exclusively affecting contaminant mass and not concentrations was the use of different gauging systems at G1 and G2. Different shapes of the measurement cross-section (triangular at G1 and rectangular at G2) caused G2 to be less precise and water imbalances on the event scale, particularly when flow was low. Summarizing the setup constraints above, we have high confidence that the experimental setup produced realistic chemograph shapes and captured peak concentration reasonably well, but are less confident regarding contaminant loads." (Lines 524-547)

**Scientific significance:**

The issues listed above reduce the scientific significance of the manuscript. Additionally, there are questions about the scientific insight conveyed by the manuscript. There are two major issues:

**Relevance of the shape of the chemograph** The effect of the chemograph is almost a trivial finding. Given the short residence time in the wetland (about 1 h; see L. 81) degradation processes will be very limited and the main effect on peak concentrations in the outlet is expected by dispersion (as expected by the authors, see below), i.e. mixing water of different concentrations at the inlet. Obviously, the more variable the input, the larger the relative effect of mixing on the relative maximum concentrations. Actually, the authors have stated this outcome already in the Introduction as fact: "'*Peak concentration reduction will be stronger for a signal with pronounced peak and low background than for a signal with small peak and high background if both signals are exposed to the same dispersion.*"' (L. 54 - 55).

[... For mathematical derivation by referee #1 see original comment ...] Reply: We agree that the finding that different signals are affected differently by dispersion is not novel and we do not claim it to be so. However, the shape of the input chemograph is usually not considered in studies on contaminant mitigation in studies. Our data show that signal shape is an important explanatory variable when assessing the functionality of VTSs, particularly when residence time is low and most of the mitigation in VTSs is due to dispersion. We made this point clearer by revising the definition of the research gap in the introduction:

"Regardless of whether VTSs target concentration reduction or mass removal, mitigation efficiency is usually associated with physicochemical properties of target compounds (Vymazal and Březinová, 2015) or VTSs, including their operation mode (Tournebize et al., 2017). However, following the concept of advective-dispersive transport (Fischer et al., 1979), the mitigating effect of dispersion on a concentration signal does not only depend on the magnitude of dispersion but also on the shape of the signal. Peak concentration reduction will be stronger for a signal with a pronounced peak and low background than for a signal with a small peak and high background if both signals are exposed to the same dispersion. Chemograph shapes, in turn, are dictated by processes in the contributing catchments. The influence of this chain of effects on contaminant mitigation and hence VTS efficiency has not been systematically investigated so far." (Lines 65-73)

In this context I am furthermore surprised that the authors did not use the model that was developed for the study wetland for evaluating the effect of different chemograph shapes (Schuetz et al., 2012). Although the wetland may have undergone changes since the last tracer experiments, it would be a useful null model for testing how different input signals influence the reduction in peak concentrations.

Reply: We thank referee #1 for this suggestion and agree that a proper transport model would be useful. In fact, we experimented quite a lot with possible representations of the investigated system in the solute transport model (OTIS) used by Schuetz et al. (2012). Although we were actually able to reproduce concentrations at the basin outlet from those at the inlet acceptably well, we decided to not include the model for the following reasons:

- (1) For evaluation of different input signals, it is crucial that solute transport (incl. dispersion) was adequately parameterized. OTIS was designed for stationary flow-conditions and as such used by Schuetz et al. (2012). Application of OTIS during transient flow is technically possible. However, we consider the parameterization in such cases questionable as model parameters that are obviously time-variable (or discharge-variable) have to be assumed constant, e.g. storage zone area, dispersion coefficient, and exchange rate.
- (2) Parameterization of transport clearly influenced model parameters related to other processes, i.e. rates of decay and kinetic sorption, so that the latter could hardly be estimated from the model.
- (3) Comparison with Schuetz et al. (2012) was not possible because their model was based on stationary discharge, while we were dealing with event data. In addition, the studied system was fundamentally changed by the implementation of the retention pond in 2016, between the experiments by Schuetz et al. (2012) and the start of sampling for the present study.

Therefore we came to the conclusion that the use of an OTIS-type model and the interpretation thereof would rather introduced additional uncertainties and that further insights provided by the model would be limited.

How to generalise the findings? It is difficult to generalise any of the findings reported in the manuscript because they are very context dependent and results such as the cluster analysis are very phenomenological. The aspect that the shape of the chemograph has an influence on the relative degree by which peak concentrations are reduced is quite evident for the type of system under investigation.

Reply: We agree that findings from field experiments often depend on local conditions. However, regarding processes in the catchment, we do not consider the results of the cluster analysis purely "phenomenological". While the results of the cluster analysis are only valid in the investigated system, we see a potential for application of this method in other systems and added this aspect to the discussion:

"..., the cluster analysis helped to explore how the catchment and processes therein influenced concentration signals of mobilized contaminants. Particularly, the analysis helped to understand under what conditions and for which pollutant sharp-peaked chemographs, associated with high acute toxicity, can be expected. We therefore see a high potential of this type of analysis for the identification of influential factors for contaminant mobilization in other catchments, although these factors may not be universal but catchment-dependent." (Lines 604-608)

**Detailed comments:**

L. 30: The phrase "'... which may be equally or more mobile, persistent and toxic than their PC ..."' is misleading because it does not mention the general case that transformation products are less toxic.

Reply: This sentence was revised and now reads:

"If degradation is incomplete, pesticides form transformation products (TPs) which in some cases may be equally or more mobile, persistent or toxic than their PCs (Hensen et al., 2020)." (Lines 39-41)

**L. 93: How often were grab samples taken?**

Reply: We clarified the number of grab sample collections:

"Pesticide monitoring at G1 and G2 consisted of 5 manual sample collections during stationary flow conditions and 10 automated event samplings during discharge events." (Lines 148-150)

L. 123: How adequate is it to only take one single isotope-labelled internal standard not corresponding to the target compounds? Checking these compounds in one of our current analytical methods, retention times vary between 16.7 (metazachlor-ESA) and 21.0 min (penconazole). Also the KOC values vary by a factor of 400 between these two compounds. Please provide additional information supporting the assumption that terbutryn as an adequate internal standard (e.g., recoveries).

Reply: We added information to the description of the analytical methodology and provided a detailed assessment of analytical precision in the supplementary material:

"The following analytical methods were used for determining pesticide levels in the water samples. Analytical standards of boscalid (99.9%), penconazole (99.1%), metazachlor (99.6%), and flufenacet (99.5%) and the internal standards Diuron-D6 (99%) and Terbutryn-D5 (98.5%) already dissolved in acetonitrile (100 µg mL-1) were purchased from Sigma-Aldrich Chemie GmbH (Steinheim, Germany). Met-ESA (95%) and met-OA (98.8%) were received from Neochema (Bodenheim, Germany). Acetonitrile (LC-MS grade; VWR International GmbH, Darmstadt, Germany) was used as organic mobile phase in chromatography and for the preparation of stock solutions. Aqueous mobile phase was prepared with ultrapure water (Membra Pure, Germany; Q1:16.6 m $\Omega$  and Q2: 18.2 m $\Omega$ .

Preparation of environmental samples (approx. 1 liter) was done by filtering with a folded filter (type 113 P Cellulose  $\phi$  240 mm). Supernatant was spiked with the internal standard Diuron-D6 (10 µl of 10 mg L-1). Extraction procedure was a solid phase extraction (SPE). Cartridges (CHROMABOND® HR-X 6 mL/200 mg) were conditioned with 10 mL methanol and washed with 10 mL pure water. 90 µL of the extract were spiked with 10 µl of Terbutryn-D5 as an internal standard. Each sample was a double determination. Measurements of environmental samples were conducted with a Triple Ouadrupole (Agilent Technologies, 1200 Infinity LC-System and 6430 Triple Quad, Waldbronn, Germany). Mobile phases were 0.01% formic acid (A) and acetonitrile (B) with a flow of 0.4 mL min-1. Gradient was as follows: 0-1 min (10% B), 1-11 min (10-50% B), 11-18 min (50-85% B), 18-21 min (85-90% B), 21-24 min (90% B), 24-26 min (90-10% B) and 26-30 (10% B). A NUCLEODUR® RP-C18 (125/2; 100-3 µm C18 ec) column (Macherey Nagel, Düren, Germany) was used as stationary phase with a set oven temperature of  $T = 30^{\circ}C$ . Calibration curve were prepared in pure water. The linearity was evaluated by preparing three curves with ten calibration points in the range 1 - 500  $\mu$ g/L. The standard curves were then extracted according to the protocol and analyzed using LC-MS/MS. The calculated linear regression values ( $R^2$ ) were very good with  $R^2$ -values > 0.999. The linearity between peak area and concentration of substances were obtained in a range of 0 - 5  $\mu$ g L-1. Hence limits of detection (LOD) and quantitation (LOQ) were calculated with DINTEST (2003) according to DIN 32645 considering an enrichment factor of 5000. LOD and LOO amounted to 0.4 and 1.3 ng  $L^{-1}$ (boscalid), 0.3 and 0.9 ng  $L^{-1}$  (penconazole), 0.3 and 1.2 ng  $L^{-1}$  (metazachlor), 0.4 and 1.3 ng  $L^{-1}$ (flufenacet) as well as 0.6 and 2.2 ng  $L^{-1}$  (met-ESA) and 0.5 and 1.6 ng  $L^{-1}$  (met-OA) considering an enrichment factor of 5000. A detailed analysis of measurement precision can be found in the *supplementary material.* " (Lines 178-207)

**L. 129 - 130: Please check the correct number of significant digits (can you measure with a precision of tens of picograms per litre?).**

Reply: The high number of digits resulted from the calculation of LODs and LOQs. We rounded these values so that they corresponded to the analytical precision (see above).

**L. 142: Please provide the version of R. I assume that you did not implement the algorithm but used kmeans() implemented in standard R.**

Reply: The version of R was 3.6.1 and the algorithm used is specified:

"The analysis was done using the software R (R Core Team, 2019) (version 3.6.1) using the 'pam' (partitioning around medoids) function from the 'cluster'-package (version 2.1.0) (Maechler et al., 2019)." (Lines 219-220)

L. 174 - 176: Why did you include DT50 a priori? I'd recommend to leave it in. The subsequent analysis would reveal whether or not is had any statistical relevance.

Reply: The linear regression model was removed. See response to referee #2.

**L. 179: How did you quantify the water balance error? Please explain.**

Reply: The equation is given in the revised manuscript:

 $"W_B = \frac{Q_{in,mean} - Q_{out,mean}}{Q_{in,mean}} \cdot 100 \%, \qquad (4)$

where  $Q_{in,mean}$  and  $Q_{out,mean}$  are the discharge at G1 and G2, respectively, averaged over the duration of the sampling procedure at both gauges.  $W_B$  was positive, if more water entered the wetland than left the wetland during the sampling procedure, and negative in the opposite case." (Lines 259-262)

L. 190: The definition of outliers and their handling is not sound. Cook's distance is simply a mean of identifying data points that deviate in a statistical sense from the rest of the data population and have a strong influence on a derived regression model. This does not imply that the data point corresponds to an outlier that can be discarded from the analysis! It may be that the outlier reflects reality as well as all other data. They may simply reflect rare events. Of course it is important to assess the influence of statistical outliers on model performance and predictions. However, unless there are sound reasons to exclude data as outliers because these reasons indicate the outliers to be wrong, outliers have to be included in the analysis. For example, it can be made transparent that some data (explicitly shown) deviate from the others in a specific way and discuss possible reasons. Hiding them to the scientific community introduces bias.

Reply: As sated above, we excluded the regression model and hence also the exclusion of outlyers.

**Response to comments by Referee #2**

The paper presents an interesting case study of the effect of vegetated treatment system to reduce contaminant inputs to surface waters. The topic is important, the field and lab work involved a lot of effort and the analysis can help to improve the understanding of contaminant loss and mitigation with vegetated treatment system. Therefore, it is worth publishing after additional clarification and correction. Overall, I have similar issues with the data analysis in the current state as reviewer 1. I support his remarks and think that they are very well stated. The main points to address as described below.

- The uncertainty in the target variables peak-concentration reduction rate and mass removal rate are very high and largely ignored in the analysis. Some reasons for the uncertainties are:
  - The contaminant concentrations are highly variable and the dynamics are difficult to capture with the applied measurement resolution. Important peaks can be missed and last data points do not always reflect baseflow concentrations. This missing information can lead to large errors in the peak-concentration reduction rates. Simple linear interpolation of concentration for mass calculation (L 157) can led to even higher bias for the mass removal rates. A flow proportional measurement could have been a better option. Was flow proportion measurements available?

Reply: We adopted our measurement setup to inculde both aspects of contaminant mitigation, i.e. peak concentration decrease and mass removal and we performed a cluser analysis to group chemograph shapes. This required measurements of original contaminant concentrations. We agree that for mass removal flow proportion sampling schemes are more accurate. We included a section in the discussion dealing with the measurement set up and associated uncertainties:

"Regarding chemographs and calculation of  $R_c$ , uncertainties arose from timing and frequency of sampling and analytical error, and additionally from discharge measurement when calculating masses and  $R_M$ . Analytical methods used in this study usually produced very consistent results so that variability in concentrations of parent compounds in duplicate samples was low (sd < 10 %). However, in individual samples collected at G1 analytical variability was elevated for met-ESA and met-OA (Figure 3), reducing confidence in concentrations and the derived measures  $R_c$  and  $R_M$  of TPs in the affected chemographs (E2, E5, E7, E8, E9). Uncertainty related to timing and frequency of sampling can hardly be quantified but certainly depends on how well the sampling intervals captured variability in concentrations during flood events and how well the time lag between upstream and downstream sampling matched the residence time of solutes in the wetland. Lefrancq et al. (2017) assessed the effect of sampling frequency in pesticide monitoring data collected during runoff from a single vineyard and found that acute toxicity of pesticide flushes was underestimated up to 4-times when calculated from event means and up to 30-times when calculated from random samples. Although these data were collected on the plot scale and we assume that variability in our catchment is lower due to longer flow paths and mixing processes on the catchment scale, uncertainty of the chemographs in our study could have been reduced by increasing sampling frequency. Regarding the timing of upstream and downstream sampling, there is evidence that water residence time in the wetland was in fact shorter than one hour. The observation that for quickly responding compounds, such as boscalid, concentration in the first sample at G2 was often elevated compared to the first sample at G1 indicates that the contaminant flush hat already reached G2 when sampling started. This did not influence determination of  $C_{out max}$  and  $R_{c}$  in the outlet of the wetland, as concentrations were still rising from the first to the second sample (Figure 3). However, effects on Mout were higher, since a relevant fraction of contaminant mass leaving the wetland was not registered and thereby caused overestimation of  $R_M$  (Figure S1). Another source of uncertainty exclusively affecting contaminant mass and not concentrations was the use of different gauging systems at G1 and G2. Different shapes of the measurement cross-section (triangular at G1 and rectangular at G2) caused G2 to be less precise and water imbalances on the event scale, particularly when flow was low. Summarizing the setup constraints above, we have high confidence that the experimental setup produced realistic chemograph shapes and captured peak concentration reasonably well, but are less confident regarding contaminant loads." (Lines 524-547)

 The timing of the last application of the investigated substances is completely neglected. However, more pronounced peaks are to be expected shortly after the application. Standardizing the concentrations or considering relative reduction rate does not completely solve this issue.

Reply: We agree that detailed information on application rates would be beneficial for interpretation of the pesticide signals emerging from the catchment. We do not have data on timing and applied amounts, but included this aspect in the discussion:

"Cluster C was composed of chemographs of all compounds but mainly from events E2 and E4 indicating event dependence. Two aspects were found to support this idea. First, there was a secondary discharge peak in event 2 that did not contribute much in terms of contaminant concentration but rather caused dilution and produced particularly flat chemograph tails. Second, peaks of herbicides and TPs were less delayed compared to fungicides. This may be the result of recent herbicide application and active surface runoff in the flat valleys. Timing of pesticide application was identified as the main driver of pesticide export by Imfeld et al. (2020) who performed a cluster analysis on rainfall data from a headwater vineyard catchment. Based on the magnitude of discharge and amount of mobilized contaminants (concentration of metazachlor  $\approx$

10  $\mu$ g L-1), both explanations seem plausible in event E2. Event E4, however, did not show particularly high herbicide concentration nor a secondary discharge peak. Although it is obvious that chemograph shapes in cluster C differed from the other clusters, unfortunately, the responsible factors remain unclear." (Lines 584-593)

And also in lines 599-603:

"The unclear interpretation of cluster C suggest that we missed important factors for the formation of chemographs. In fact, variables like spatial distribution of rainfall or pesticide application rates and timing (Imfeld et al., 2020) and possibly other factors likely influenced chemograph shapes. Knowing all these variables would not change the results produced by the clustering algorithm but rather increase our ability to interpret them. Nevertheless, the cluster analysis helped to explore how the catchment and processes therein influenced concentration signals of mobilized contaminants." (Lines 601-608)

I think a much better understanding of the uncertainty of the contamination measurement could be gained from a detailed analysis of the discharge behavior during the investigated events. This data is available in a much higher resolution (L86: Stream flow was measured every minute ...). Unfortunately, they are neither shown in detail nor really used in the analysis. It would be interesting to see the sample points/concentrations during an event together with the discharge measurement on a higher resolution, since the dynamics of the contaminants are driven mainly by the hydrology. I think this additional information would give an inside about how well the concentration patterns would improve the interpretation as well.

Reply: We added a Figure (Figure 3) showing discharge dynamics and pesticide responses in detail (see above) and included discharge dynamics in the discussion:

"Cluster D included chemographs of both herbicides and TPs and presented a clear peak that was often defined by a single sample 2 h after the beginning of the event. In contrast to cluster A, cluster D was characterized by a single sharp discharge peak (except in event E7 where a second peak occurred shortly after the first) and mainly included chemographs during periods of low flow. Our interpretation is that cluster D represented flow events in which no dilution of herbicide and TP fluxes by fungicide fluxes or secondary discharge peaks occurred. Low pre-event discharge in cluster D compared to cluster A may indicate low water levels which may have caused a slower response as no enriched pre-event water was released from the soils in the valleys. " (Lines 594-600)

- The regression analysis is done rather poorly and the procedure is neither well explained nor well presented.
  - It is not shown that the condition for a simple multiple linear regression are fulfilled, since the results are not validated or at least this information is not shown. I would at least expect a classical residual analysis in the supporting information.

Reply: We are thankful for this concern. After careful consideration of the statistical constraints of our data set, particularly with respect to the following points, we decided to replace the regression analysis by a qualitative comparison of mitigation in the wetland to compound properties, discharge dynamics and chemograph shape parameters. The results are shown in Figure 5 and Figure 6 and discussed as follows:

**"4.3 Mitigation efficiency and chemograph shape**

**4.3.1 Peak concentration reduction**

We hypothesized that peak concentration reduction in the VTS will be highest for chemographs with the sharpest peaks, i.e. for the chemographs that were most sensitive to dispersion. And indeed we found a systematic relationship between  $R_C$  and both  $i_{DS}$  and chemograph clusters. Although the relationship of clusters and  $R_C$  largely reflected the relationship between  $R_C$  and  $i_{DS}$ , it is surprising that  $R_C$  was clearly highest in cluster D and not in cluster C which presented better defined peaks and slightly higher iDS per cluster (Figure 4f). Critical inspection of input chemographs shows that in several chemographs of TPs (met-ESA and met-OA in event E4 and met-OA in event E8) elevated concentrations in the last samples exhibited high analytical errors and did not appear in the outlet chemograph. These dubious samples caused low  $i_{DS}$  but substantial  $R_C$  and thus contributed to variability in  $i_{DS}$  despite high values of  $R_C$  in cluster D. We therefore do not consider the deviation from the expected cluster ordering contradictory but to result from increased uncertainty in cluster D as mentioned earlier. In contrast, the hypothesized relationship between RC and chemograph shape was demonstrated for both iDS and chemograph clusters, the latter of which also integrates shape aspects that go beyond iDS, e.g. timing of peaks. Overall, the values of  $R_{\rm C}$  found in our study compare with field data from vegetated buffers (Bundschuh et al., 2016; Stehle et al., 2011) and are in the range of those found in vegetated stream mesocosms by Elsaesser et al. (2011) and Stang et al. (2014) who both attributed most of the observed peak reduction to dispersion.

In addition, we found relationships between and  $R_C$  and discharge dynamics, i.e.  $Q_{mea}$  and ratio of  $Q_{max}$  to  $Q_{mean}$ . The influence of discharge on  $R_C$  may be two-fold. First, increasing flow reduced residence time and hydraulic efficiency, i.e. short circuiting reduced the potential for dispersion and interaction with wetland sediments or plants. Second, the fact that chemographs of events with high  $Q_{max}$  to  $Q_{mean}$  ratios were attributed to cluster D suggests that discharge dynamics influenced the shape of the chemograph at the wetland inlet. This means, the influence of discharge may also be indirect by promoting the formation of sharp-peaked chemographs with high potential for peak reduction.

In contrast to other studies, we did not find clear relationships of  $R_C$  to and physiochemical properties of compounds such as sorption affinity (Stehle et al., 2011; Vymazal and Březinová, 2015) or solubility (Bundschuh et al., 2016). The absence of such relationships may partially be due to the low number of different target compounds in our study (n=6). However, given the short time lag between sampling at the inlet and outlet of the wetland ( $\Delta t = 1h$ ), it seems logical that no relevant sorption or degradation occurred within this period. For comparison, in batch experiments by Gaullier et al. (2018) adsorption equilibrium for boscalid (compound with second highest Kfoc in our study) was only reached after 24 h. Despite the relatively narrowly confined  $R_C$  values of the parent compounds, we do not consider physiochemical compound properties as major drivers of  $R_C$  in our VTS.

**4.3.2 Contaminant mass removal**

For *RM* we found a different pattern among the chemograph clusters than for *RC*. *RM* was apparently higher in clusters A and D than in clusters B and C. However, the clusters indicating substantial mass removal were those with increased uncertainty regarding compound mass. Cluster A often showed relevant mass flux at the end of sampling (and presumably beyond) which we did not account for. Cluster D contained dubious data points of TPs and poorly defined peaks outside the periods of high sampling frequency. In addition, due to overestimation of solute travel time in the wetland in the monitoring setup, the rising limp of the mass flux signal at G2 was often not adequately captured by the sampling scheme, causing underestimation of downstream event mass and overestimation of mass loss. In absence of any clear relationship with compound properties, discharge dynamics or chemograph shape, this suggests that the assessment of contaminant masses was subject to systematical errors and that the apparent mass loss found in our study should therefore not be over-interpreted.

In earlier studies, Lange et al. (2011) and Schuetz et al. (2012) observed a 15-30 % mass loss of the fluorescent tracer sulforhodamine-B in the wetland subsection of the current VTS. These results indicate a general potential for sorption of organic compounds in this system, but represent an earlier succession state of the wetland and stationary flow conditions with much longer residence times. Also in the current VTS kinetic sorption of contaminants may have occurred but sorption equilibrium was certainly not reached (Gaullier et al., 2018). Thus the effect of sorption did not reach its full potential. In fact, other studies reported limited mass removal in wetlands with comparable residence times. Ramos et al. (2019) did not find relevant  $R_M$  in two surface flow wetlands with residence times between 45 min and 6 h in England. In contrast, Passeport et al. (2013) found  $R_M$  between 45 % and 96 % in a constructed wetland with a residence time of 66.5 h. However, their contaminant mass loss coincided with loss of water (45 %). Mesocosm experiments by Elsaesser et al. (2011) and Stang et al. (2014) showed strong concentration reduction but only very limited and temporary mass removal at residence times of a few hours. In summary, these findings suggest that the potential for mass removal in wetland systems like the one studied here is rather limited. However, wetlands have been shown to reduce contaminant mass, when residence times are sufficiently long (Gregoire et al., 2009) or when operated in batch mode (Tournebize et al., 2017; Moore et al., 2000; Maillard et al., 2016)." (Lines 609-661)

 Automatically remove outliers based on a doubtful model without further analysis is not a proper way to go. For example, if outliers are a real problem, robust regression could be a solution (e.g. library robustbase).

Reply: As stated above, we entirely refrained from removing outliers in the revised manuscript.

I don't think the requirement of independence of the data is fulfilled in this context. Data points of the same discharge event for the different components are not expected to be independent. Maybe a mixed model (with the discharge event as random effect) could help (e.g. library Ime4). However, I doubt that more than a nice qualitative analysis of contaminate dynamics in a catchment with a vegetated treatment system will be possible with this setup.

Reply: See also above. We are grateful for this comment and the helpful suggestion to perform a qualitative analysis instead of a questionable regression model.

 It is somehow obvious that dispersion has a stronger effect on substances with a more pronounced peak (as explained by reviewer 1).

Reply: We agree with referee #2 that it is obvious that dispersion affects sharp peaks stronger than flattened signals. However, as already discussed in our response to reviewer #1, it is not adequately stressed in existing literature on wetland contaminant mitigation.

Although the clustering is done correctly, the connection with the discharge events is not well elaborated. Moreover, they are other clustering algorithms, which might be more robust (e.g. k-medoids. hierarchical Clustering). In L 207 it is written: "With the exception of cluster B which rather represented similar events (event 1 and event 4 in Fig. 2), overall clustering was controlled by similar behavior of contaminant groups." What was special by the event 1 and 4? Are these really exceptions? The contaminant groups seem to be important, however, I think the discharge dynamic and the application timing are important as well. Maybe it would also been interesting to cluster the discharge events. This data are also available in a higher resolution.

Reply: We changed the clustering algorithm to k-medoids which should increase robustness against outliers:

"Identification of patterns in input chemographs was done by k-medoids cluster analysis - a variation of the commonly applied k-means algorithm. Both approaches partition the elements of a dataset into a predefined number k of clusters by attributing the elements to the cluster with the nearest cluster center. Optimal clustering is achieved by iteratively updating cluster centers and minimizing distance between data points and cluster centers. K-medoids differs from k-means as it uses existing points (medoids) as cluster centers instead of means and is considered more robust against extreme values and outliers (Han et al., 2012). A total of 58 concentration sequences was included in the analysis, consisting of 10 sequences per target compound, except for flufenacet which did not exceed LOQ in two events. Prior to cluster analysis, data was normalized by the maximum of each chemograph to promote that clustering represented shape, rather than differences in absolute concentration. The analysis was done using the software R (R Core Team, 2019) (version 3.6.1) using the 'pam' (partitioning around medoids) function from the 'cluster'package (version 2.1.0) (Maechler et al., 2019). We tested clustering for k ranging between 2 and 10, the final number was determined by both visual inspection of the clusters and assessment of explanatory benefit per additional cluster (elbow method). As a result we found that k=4 resulted *in the best partition.* " (Lines 211-222)

Change of clustering algorithm caused minor differences in attribution of chemographs to clusters as described by Figure 4 and the following paragraph:

"Cluster A (Figure 4) was characterized by absence of a clear peak during the first two hours of sampling but elevated concentrations during later times, resulting in low  $i_{DS}$ . Cluster B showed a quick response, i.e. concentrations increased sharply within the first 30 minutes. Concentrations were the highest of all clusters and still elevated in the last sample compared to pre-event levels. Cluster C was characterized by a clear peak within the first two hours and a low tailing and was

the cluster with highest median iDS. Cluster D showed the most inconsistent pattern and maximum concentrations appeared later compared to clusters B and C. A relatively clear pattern was evident in the attribution of compounds to the clusters. Chemographs of the fungicides boscalid and penconalzole were mainly assigned to cluster B, while the herbicides and the TPs were assigned to the remaining three clusters. Cluster A was composed of herbicide and TP chemographs, particularly from events with multiple discharge peaks. Cluster D represented chemographs of herbicides and TPs mainly during the events E5 to E8 which were all characterized by sharp discharge peaks during periods of generally low flow (Figure 3). Almost all chemographs of the events E2 and E4 were attributed to cluster C." (Lines 343-353)

We put more emphasis on the discussion of chemograph clusters, including the role of discharge conditions, and particularities of certain events:

"Despite highly variable application patterns, our cluster analysis resulted in four groups with similar chemograph shape. Many factors have been shown in literature to influence the mobilization of pesticides in catchments, including catchment properties, event properties and physiochemical compound properties. As catchment properties we here consider factors associated with runoff generation such as catchment geometry, terrain slopes, and in particular the delineation of areas where different compounds were applied. The interplay of these factors defines hydrological activity and connectivity (i.e. by shortcuts like roads and drainage pipes) of critical source areas for different compounds (Doppler et al., 2012; Gomides Freitas et al., 2008). Event properties include intensity and dynamics of rainfall (Imfeld et al., 2020) and subsequent runoff (Doppler et al., 2014). Relevant physiochemical compound properties are e.g. mobility and degradability (Gassmann et al., 2015).

These properties are reflected to varying degrees in the results of the cluster analysis. Cluster A was characterized by a quick response and a concentration plateau towards the end of sampling and was mainly composed of TPs. The fact that concentration maxima in cluster A were retarded compared to fungicides (cluster B), although their parent compounds were applied closer to the stream in the flat valley bottoms, suggests that they were transported with a slower flow components. Due to flatter terrain, surface runoff played a less important role and the main transport pathway was subsurface flow. Where fields were undrained, however, transit time of water from the infiltration point to the stream from the fields in the valley during discharge events would therefore be pre-event water, enriched in TPs formed in the soil, corresponding to the formation site of TPs of the chloracetamide herbicides to which metazachlor belongs (Mersie et al., 2004). Seepage of pre-event TP-rich water thus explains the immediate response of chemographs in cluster A. The quick response was often followed by a local concentration minimum between samples 2 and 5, i.e. between 30 min and 6 h after sampling was initialized.

Coincidence of this minimum with concentration peaks of fungicides might suggest dilution of TP concentration by mixing with event water carrying high loads of fungicides but less TPs of metazachlor.

Cluster B represented differences between fungicides and the remaining compounds. Considering land use distribution in the studied catchment, it is unclear whether this partition reflects different compound properties or catchment properties or both. The fact that concentration in cluster B quickly increased with discharge (within 30 minutes) is in line with fast transport from the vineyard terraces to the stream via roads and drainage pipes as described by Gassmann et al. (2012) for suspended solids in the studied catchment. Along such preferential pathways, compound properties, such as sorption affinity, may be less important (Gomides Freitas et al., 2008) compared to e.g. percolation through the soil with intense contact to sorption sites in the soil matrix. Moreover, fungicides are applied by sprayers into the foliage and can drift to e.g. paved surfaces from which they can be quickly mobilized by subsequent rainfall (Lefrancq et al., 2013). We therefore hypothesize that cluster B was mainly produced by surface flushing and fast transport pathways of fungicides. This explained the quick rise and subsequent decline in concentrations (concurrent with plateaus produced by slower flow components in cluster A).

Cluster C was composed of chemographs of all compounds but mainly from events E2 and E4 indicating event dependence. Two aspects were found to support this idea. First, there was a secondary discharge peak in event 2 that did not contribute much in terms of contaminant concentration but rather caused dilution and produced particularly flat chemograph tails. Second, peaks of herbicides and TPs were less delayed compared to fungicides. This may be the result of recent herbicide application and active surface runoff in the flat valleys. Timing of pesticide application was identified as the main driver of pesticide export by Imfeld et al. (2020) who performed a cluster analysis on rainfall data from a headwater vineyard catchment. Based on the magnitude of discharge and amount of mobilized contaminants (concentration of metazachlor  $\approx$  10 µg  $L^{-1}$ ), both explanations seem plausible in event E2. Event E4, however, did not show particularly high herbicide concentration nor a secondary discharge peak. Although it is obvious that chemograph shapes in cluster C differed from the other clusters, unfortunately, the responsible factors remain unclear.

Cluster D included chemographs of both herbicides and TPs and presented a clear peak that was often defined by a single sample 2 h after the beginning of the event. In contrast to cluster A, cluster D was characterized by a single sharp discharge peak (except in event E7 where a second peak occurred shortly after the first) and mainly included chemographs during periods of low flow. Our interpretation is that cluster D represented flow events in which no dilution of herbicide and TP fluxes by fungicide fluxes or secondary discharge peaks occurred. Low pre-event discharge in cluster D compared to cluster A may indicate low water levels which may have caused a slower response as no enriched pre-event water was released from the soils in the valleys.

The uncertain interpretation of cluster C suggest that we missed important factors for the formation of chemographs. In fact, variables like spatial distribution of rainfall or pesticide application rates and timing (Imfeld et al., 2020) and possibly other factors likely influenced chemograph shapes. Knowing all these variables would not change the results produced by the clustering algorithm but rather increase our ability to interpret them. Nevertheless, the cluster analysis helped to explore how the catchment and processes therein influenced concentration signals of mobilized contaminants." (Lines 522-608)

**Detailed comments**

L100: Fig. 2: I guess the discharge shown in Fig. 2 is from G1. This should be included in the description. Reply: We replaced Figure 2 by a more detailed version (Figure 3) with a proper legend (see above).

L 108: Overall, herbicides have been also shown to be very persistent. For examples, atrazine has been detected after 10 years without application.

Reply: This was not meant to be a general statement. These lines (also s. next comment) just sum up, how the selected compounds are classified according to the Pesticide Properties Data Base (Lewis et al., 2016). We clarified this in the revised manuscript:

"In this study, we focused on 6 target compounds including the fungicides boscalid and penconazole, the herbicides metazachlor and flufenacet, and the TPs metazachlor sulfonic acid (met-ESA) and metazachlor oxalic acid (met-OA). Selected physicochemical properties of the target compounds are listed in Table 1. According to the Pesticide Properties Data Base (Lewis et al., 2016) the contaminants can be classified as low (boscalid) to moderately soluble in water, very mobile (TPs) to slightly mobile (fungicides). The target fungicides are considered moderately fast degradable in the water phase and persistent in soils, while the target herbicides are considered stable in the water phase and non-persistent in soils. TPs of metazachlor are considerably more persistent in soil than their PC. The fungicides are considered stable with respect to hydrolysis but degradable via photolysis, while the herbicides are stable regarding both." (Lines 130-137)

L 110: Azole pesticides are also persistent as indicated by many studies (e.g. https://doi.org/10.1016/j.envint.2020.105708) Reply: (s. above)

L 131: What is the accuracy and precision of the method? Has the analytical method validated? Reply: We provided additional information about the analytical procedure in the method section and added a detailed analysis of the measurement precision in the supplementary material. "The following analytical methods were used for determining pesticide levels in the water samples. Analytical standards of boscalid (99.9%), penconazole (99.1%), metazachlor (99.6%), and flufenacet (99.5%) and the internal standards Diuron-D6 (99%) and Terbutryn-D5 (98.5%) already dissolved in acetonitrile (100 µg mL-1) were purchased from Sigma-Aldrich Chemie GmbH (Steinheim, Germany). Met-ESA (95%) and met-OA (98.8%) were received from Neochema (Bodenheim, Germany). Acetonitrile (LC-MS grade; VWR International GmbH, Darmstadt, Germany) was used as organic mobile phase in chromatography and for the preparation of stock solutions. Aqueous mobile phase was prepared with ultrapure water (Membra Pure, Germany; Q1:16.6 m $\Omega$  and Q2: 18.2 m $\Omega$ .

Preparation of environmental samples (approx. 1 liter) was done by filtering with a folded filter (type 113 P Cellulose  $\phi$  240 mm). Supernatant was spiked with the internal standard Diuron-D6 (10 µl of 10 mg L-1). Extraction procedure was a solid phase extraction (SPE). Cartridges (CHROMABOND® HR-X 6 mL/200 mg) were conditioned with 10 mL methanol and washed with 10 mL pure water. 90 µL of the extract were spiked with 10 µl of Terbutryn-D5 as an internal standard. Each sample was a double determination. Measurements of environmental samples were conducted with a Triple Quadrupole (Agilent Technologies, 1200 Infinity LC-System and 6430 Triple Quad, Waldbronn, Germany). Mobile phases were 0.01% formic acid (A) and acetonitrile (B) with a flow of 0.4 mL min-1. Gradient was as follows: 0-1 min (10% B), 1-11 min (10-50% B), 11-18 min (50-85% B), 18-21 min (85-90% B), 21-24 min (90% B), 24-26 min (90-10% B) and 26-30 (10% B). A NUCLEODUR® RP-C18 (125/2; 100-3 µm C18 ec) column (Macherey Nagel, Düren, Germany) was used as stationary phase with a set oven temperature of  $T = 30^{\circ}C$ . Calibration curve were prepared in pure water. The linearity was evaluated by preparing three curves with ten calibration points in the range 1 - 500  $\mu$ g/L. The standard curves were then extracted according to the protocol and analyzed using LC-MS/MS. The calculated linear regression values ( $R^2$ ) were very good with  $R^2$ -values > 0.999. The linearity between peak area and concentration of substances were obtained in a range of 0 - 5  $\mu$ g L-1. Hence limits of detection (LOD) and quantitation (LOQ) were calculated with DINTEST (2003) according to DIN 32645 considering an enrichment factor of 5000. LOD and LOO amounted to 0.4 and 1.3 ng  $L^{-1}$ (boscalid), 0.3 and 0.9 ng  $L^{-1}$  (penconazole), 0.3 and 1.2 ng  $L^{-1}$  (metazachlor), 0.4 and 1.3 ng  $L^{-1}$ (flufenacet) as well as 0.6 and 2.2 ng  $L^{-1}$  (met-ESA) and 0.5 and 1.6 ng  $L^{-1}$  (met-OA) considering an enrichment factor of 5000. A detailed analysis of measurement precision can be found in the *supplementary material.* " (Lines 178-207)

L 145: Why is the cluster analysis important for the calculation of the dispersion sensitivity index? The index could also be calculated without clustering.

Reply: We acknowledge that this phrasing was imprecise. We made the connection clear in the revised manuscript.

L 191: From which mean? Do you mean 2 standard deviation from the prediction? Reply: No outliers were removed in the revised manuscript.

L 205: It doesn't make sense to talk about a peak in Cluster C ("T peak = 6h"). Not even the mean has a peak there.

Reply: This is true and was avoided in the revised manuscript.

L 212: The surface runoff from the elevated vineyard has also to flow through the lower terrain slope to reach the river, expect that there are other shortcuts (streets, drains). See also reviewer 1).

Reply: We thank referee #2 for this comment and put stronger emphasis on the role of artificial drainage and other shortcust inside the catchment in the revised manuscript both in the description of the study site:

"Large parts of the catchment are drained by a sub-surface pipe network (Figure 1) connecting vineyards and paved roads to the main channel in the valley. This drainage network causes fast downstream transport of storm water and suspended sediments (Gassmann et al., 2012). In addition, fields in the valley bottoms are drained by a secondary network of smaller and usually shorter field drains that are either connected to the primary drainage network or directly connected to the stream (Schuetz et al., 2016)." (Lines 106-111)

We also included this aspect in the discussion of contaminant mobilization, particularly in the case of cluster B:

"Cluster B represented differences between fungicides and the remaining compounds. Considering land use distribution in the studied catchment, it is unclear whether this partition reflects different compound properties or catchment properties or both. The fact that concentration in cluster B quickly increased with discharge (within 30 minutes) is in line with fast transport from the vineyard terraces to the stream via roads and drainage pipes as described by Gassmann et al. (2012) for suspended solids in the studied catchment. Along such preferential pathways, compound properties, such as sorption affinity, may be less important (Gomides Freitas et al., 2008) compared to e.g. percolation through the soil with intense contact to sorption sites in the soil matrix. Moreover, fungicides are applied by sprayers into the foliage and can drift to e.g. paved surfaces from which they can be quickly mobilized by subsequent rainfall (Lefrancq et al., 2013). We therefore hypothesize that cluster B was mainly produced by surface flushing and fast transport pathways of fungicides. This explained the quick rise and subsequent decline in concentrations (concurrent with plateaus produced by slower flow components in cluster A)." (Lines 574-583) L 315: I do not understand the explanatory power for the different variable. Are they calculated by a univariate analyses? At least for me, the R-Output would be much easier to interpret. Reply: The regression analysis is no longer part of the manuscript.

**Pesticide peak concentration reduction in a small vegetated treatment system controlled by chemograph shape**

3

4

Jan Greiwe1, Oliver Olsson2, Klaus Kümmerer2, Jens Lange1

5 1Hydrology, University of Freiburg, Fahnenbergplatz, 79098 Freiburg, Germany

[revised manuscript text omitted]
 hypothesize that mitigation efficiency in wetland systems with short residence time mainly depends on the chemograph
- 80 shape of the mobilized contaminants. In order to test this hypothesis, we monitored the mobilization of 6 organic contaminants
- 81 during 10 discharge events in a viticultural catchment. Then, we grouped the resulting chemographs into clusters of similar

82 shape. Finally, we compared RC and RM among clusters and assessed how they were related to other shape-related parameters, 83 discharge dynamics, and physicochemical compound properties. In this study, we We test this hypothesis in three steps. First, 84 we assess monitored the mobilization of six 6 organic contaminants during 10 discharge events in response to rainfall in a 85 viticultural catchment. Second, we grouped the resulting chemographs according to shape similarity. Third, andwe evaluated 86 the influence of the contaminant concentration signal chemograph shape on mitigation efficiencyefficiencies by comparing 87 peak concentration reduction and mass removal rates to chemograph shape parameters as well discharge dynamics and 88 physicochemical compound properties.

89

in a VTS located at the catchments outlet. We hypothesize that mitigation efficiency does not only depend on properties of
 the contaminant (e.g. sorption affinity, water and soil half lives) or the treatment system (e.g. retention time, plant coverage),

92 but also on the shape of the concentration signal mobilized in the catchment. To test this hypothesis, we

93 search for patterns in contaminant flush signals by performing a cluster analysis on flow triggered monitoring data, integrate

94 the results into two multiple linear regression models for peak concentration reduction and mass removal, respectively, and 95 evaluate the relative importance of the model variables.

**96 2 Material and methods**

**97 2.1 Study site**

98 The study site (Figure 1Fig. 1) is located inside a flood detention basin in a-the 1.8 km2 Loechernbach headwater-catchment, 99 southwest Germany. Catchment elevation ranges from between 213 m.a.s.l. at the outlet to and 378 m.a.s.l. in the western 100 corner. Mean precipitation was 800 mm a-1 between 2009 and 2018, mean air temperature was-11.3 °C. Soils mainly consist 101 of calcaric regosols which formed on Aaeolian loess and have a typical grain size distribution of 10 % sand, 80 % silt and 102 10 % clay. Most of the land in the catchmment area is dedicated to viticulture on large artificial vineyard terraces -(71 %). 103 while croplands occupy the main-valley bottoms (20%). Forest only accounts for a small portion (9%) and is limited to the 104 most elevated part of the catchment. This partition in land use is reflected in the main application areas of pesticide types. 105 Fungicides are applied to the on vineyard terraces, while and herbicides are mainly applied to the The elevated vineyard terraces 106 are subject to frequent fungicide application, while herbicides are applied to the cropland in the flat valleys. Large parts of the 107 catchment are drained by a dense-sub-surface pipe network (Figure 1) with a total length of about 9 km directly-connecting 108 vineyards and paved roads to the main channel in the valley. The his drainage network which causes fast downstream transport 109 of storm water as well as and of dissolved and suspended material sediments (Gassmann et al., 2012). In addition, fields in the 110 valley bottoms are drained by a secondary network of smaller and usually shorter field drains that are either connected to the 111 primary drainage network or directly connected to the stream (Schuetz et al., 2016). A 20,000 m3 detention basin was built at 112 the outlet of the Loechernbach to prevent flooding of the downstream village.

Inside the detention basin, a 258 m2 vegetated surface flow constructed wetland and a 105 m2 retention pond (maximum depth 1.5 m) are connected in series parallel to the course of the Loechernbach stream. <del>During baseflow conditions a</del>A small dam diverts all flow through the vegetated treatment systems during base flow conditions, but allows water to bybass the VTS

diverts all flow through the vegetated treatment systems during base flow conditions, but allows water to bybass the VTS during large discharge events. In the case of large discharge events, the treatment systems are bypassed. The wetland is in

117 operation since 2010 and its succession was studied by Schuetz et al. (2012). The pond was added to the system in January

118 2016. Both The entire detention basin is sealed by an impermeable clay layer that prevents infiltrationleakage to groundwater.

119 From previous studies, it is known that wWater residence timesHRT in the system ranges from about less than one hour during

120 flood events up to more than aseveral days during extreme low flow conditions.